Article 

# Greater wax moth control in apiaries can be improved by combining *Bacillus thuringiensis* and entrapments

Bo Han[1,7], Li Zhang [1,7], Lili Geng[2,7], Huiru Jia[2,7], Jian Wang[2], Li Ke[1], Airui Li[1], Jing Gao[1], Tong Wu[1], Ying Lu[1], Feng Liu[3], Huailei Song[4], Xiaoping Wei[5], Shilong Ma[6], Hongping Zhan[5], Yanyan Wu[1], Yongjun Liu[1], Qiang Wang[1], Qingyun Diao[1], Jie Zhang[2] & Pingli Dai [1] ✉

The greater wax moth (GWM), *Galleria mellonella* (Lepidoptera: Pyralidae), is a major bee pest that causes significant damage to beehives and results in economic losses. *Bacillus thuringiensis* (Bt) appears as a potential sustainable solution to control this pest. Here, we develop a novel Bt strain (designated BiotGm) that exhibits insecticidal activity against GWM larvae with a $LC_{50}$ value lower than 2 µg/g, and low toxicity levels to honey bee with a $LC_{50} = 20598.78$ µg/mL for larvae and no observed adverse effect concentration = 100 µg/mL for adults. We design an entrapment method consisting of a lure for GWM larvae, BiotGm, and a trapping device that prevents bees from contacting the lure. We find that this method reduces the population of GWM larvae in both laboratory and field trials. Overall, these results provide a promising direction for the application of Bt-based biological control of GWM in beehives, although further optimization remain necessary.

Honey bees are indispensable pollinators in ecosystems worldwide, underpinning biodiversity and economic growth[1–4]. However, both feral and wild honey bee populations have experienced significant declines in recent decades, raising alarms among beekeepers, scientists, and the general public[5–8]. This decline poses a significant threat to global food and nutritional security, given the significant role of bees in food production[1,9,10]. Substantial evidence suggests that the decline in bee populations may be attributed to a wide range of factors, such as pathogens, parasites, pests, predators and chemical pesticides[11–14]. Of particular concern is the greater wax moth (GWM), *Galleria mellonella*, a significant threat to bee colonies globally[15].

The GWM is a pervasive pest of honey bee colonies worldwide[16] (Supplementary Fig. 1). Its larvae exhibit a destructive feeding habit and can feed on wax combs, cast larval skins, pollen, and honey,

causing the most damage to dark and stored combs, which are preferred by the GWM[15,16]. In severe cases, if the pest invades the comb to a higher level, the colony may suffer severe damage and bees may flee the hive[16]. Additionally, studies suggest that moths can transmit viral pathogens such as Israeli acute paralysis virus (IAPV) and black queen cell virus (BQCV) to larvae[17], further emphasizing the need for effective countermeasures.

Indeed, various methods have been proposed as alternatives for GWM control, and the use of biological agents, particularly toxins from the bacterium *Bacillus thuringiensis* (Bt), has emerged as a potential solution[15,16,18]. Bt demonstrates insecticidal activity against numerous pests (>600 insect species), offering a targeted and environmentally friendly option[19–24]. Currently, Bt dominates the biopesticide market, accounting for 50% of the global market for this type of product[25].

[1]State Key Laboratory of Resource Insects, Institute of Apicultural Research, Chinese Academy of Agricultural Sciences, Beijing 100193, China. [2]State Key Laboratory for Biology of Plant Diseases and Insect Pests, Institute of Plant Protection, Chinese Academy of Agricultural Sciences, Beijing 100193, China. [3]Jiangxi Institute of Apicultural Research, Nanchang 330201, China. [4]Shanxi Agricultural University, Taiyuan 030006, China. [5]Modern Agricultural Development Institute, Guizhou Academy of Agricultural Sciences, Guiyang 550006, China. [6]Enshi Academy of Agricultural Sciences, Enshi 445002, China. [7]These authors contributed equally: Bo Han, Li Zhang, Lili Geng and Huiru Jia. ✉e-mail: daipingli@caas.cn

However, the effectiveness of Bt bioinsecticides can be influenced by various environmental factors, including the formulation type, application method, contaminants, and environmental conditions, such as ultraviolet (U.V.) radiation, temperature, pH, and rainfall[26–31]. Hence, improved formulations and applications are actively sought, with encapsulation techniques showing promise[26].

The potential of Bt against GWM has been acknowledged[15,18]. Several issues hinder its widespread adoption: (i) the toxicity and efficacy of Bt on GWM larvae have not been clearly reported due to the lack of standard bioassay methods; (ii) there is a high potential for pest resistance to develop, as Bt strains with high insecticidal activity ($LC_{50} < 5 \mu g/g$) against GWM have not been screened; (iii) the two methods that are currently commonly used, directly spraying and dipping or mixing it into the comb foundation with Bt, may result in Bt spore residues in honey, pollen, and comb, posing a potential threat to the safety of bees and their products; and (iv) although some experiments have shown higher larval mortality[18,32], field evaluations on its practicality have not been reported.

To address these limitations, the present study was conducted. More specifically, we first investigated the damage caused by the moth and the need for control in beekeeping. Moreover, we evaluated the potential insecticidal efficacy of five types of Bt against 2nd instar larvae through self-established bioassay methods and further assessed their effects by spraying them onto the comb or pressing them into the comb foundation under indoor conditions. Additionally, the safety of BiotGm (a novel *B. thuringiensis* strain with high insecticidal activity against GWM larvae and enormous potential applications in pest management) on *Apis mellifera* and *Apis cerana cerana*, the two most common honey bee species in agroecosystems in China, was comprehensively evaluated. Last, but most importantly, we developed a novel biocontrol entrapment method by discovering lures capable of trapping GWM larvae and combining them with high insecticidal activity BiotGm. The entrapment included a trapping device to prevent bees from accessing the lures. Notably, the efficacy of this biocontrol entrapment was determined through laboratory and field trials.

## Results

### Toxicity of Bt against GWM

The $LC_{50}$ values of G033A, G033A-1, KN11, KN11-1, and BiotGm to GWM larvae are tested by the self-established bioassay (Fig. 1), and the toxicity from most to least toxic was BiotGm > KN11 > G033A = KN11-1 = G033A-1 (Table 1). Notably, there was no difference ($P > 0.01$) in the toxicity of G033A, G033A-1, and KN11-1 because of the overlap of 95% confidence intervals.

*B. thuringiensis* BiotGm with the most toxicity to GWM larvae in the bioassay was observed to produce approximately bipyramidal crystals using optical microscopy and atomic force microscopy (Supplementary Fig. 2a, b). The genome of *Bt* BiotGm was sequenced and the data was deposited in GenBank under accession numbers CP130743 to CP130747. The complete genome sequence of BiotGm consists of one chromosome with a total length of 5,718,068 bases and four plasmids (Supplementary Table 1). According to the CVTree phylogenetic analysis, BiotGm belonged to subspecies *aizawai* (Supplementary Fig. 2c). A total of 6,869 coding sequences (CDS) were predicted, including 6 insecticidal crystal proteins (Cry1Aa1, Cry1Ca7, Cry1Da1, Cry1Ia10, Cry2Ab1, Cry9Ea1) and 1 vegetative insecticidal protein (Vip3Aa11) (Supplementary Table 2).

Furthermore, we performed a comparative histopathology analysis of the midgut to assess the pathological effects of BiotGm on GWM larvae. As shown in Fig. 2, the midgut treated with PBS was normal, as epithelial cells were arranged in regular lines and rested on the intact basement membrane. In addition, a clear dense chromosome structure was also observed. In contrast, BiotGm treatment negatively altered gut morphology in terms of displacement of the epithelial cell arrangement and a discontinuous gut barrier towards the body cavity 2 h after inoculation, and the midgut cells sloughed severely with increasing time. The midgut cells almost completely disappeared, and intestinal walls cracked in the severely damaged area 10 h after inoculation. Moreover, terminal deoxynucleotidyl transferase-mediated digoxigenin-dUTP-biotin nick-end labelling (TUNEL)-positive cells typically emitted green fluorescence at 488 nm, signifying apoptotic cells. A marked propensity for apoptosis was

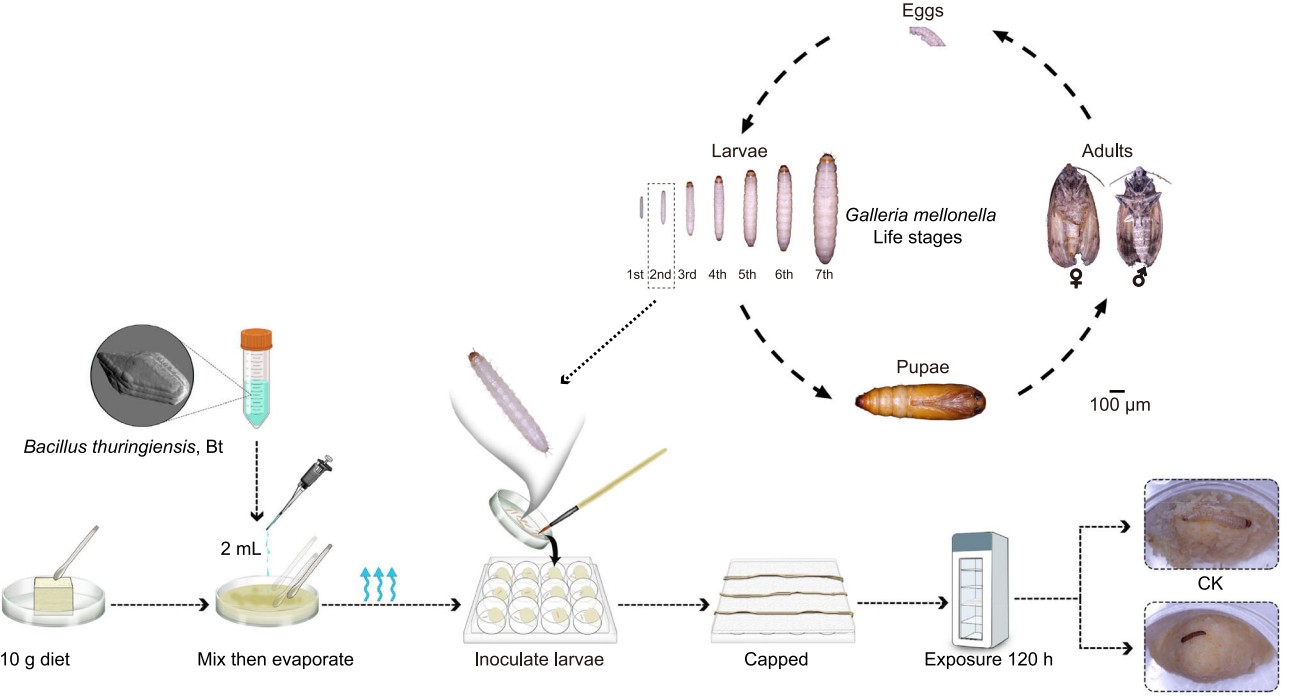

**Fig. 1 | Using the self-established bioassay method to determine acute oral toxicities of five Bt to the GWM larvae.** Representative images of the GWM at each developmental stage and the flow chart of bioassay method used in this study to determine the toxicity of Bt against the GWM larvae.

**Table 1 | Acute oral toxicities of five Bt to the GWM larvae**

| Bt | $n$ | $R^2$ | LC$_{50}$ (µg/g) [a] | Toxic regression equations |
|---|---|---|---|---|
| G033A | 288 | 0.9217 | 5.999 (3.960–9.089) | $y = 1.6084x + 3.7485$ |
| G033A-1 | 288 | 0.8962 | 10.007 (8.279–12.095) | $y = 2.5536x + 2.4457$ |
| KN11 | 288 | 0.9640 | 3.004 (2.482–3.635) | $y = 2.2824x + 3.9098$ |
| KN11-1 | 336 | 0.8083 | 7.044 (5.603–8.855) | $y = 2.4482x + 2.9244$ |
| BiotGm | 284 | 0.9730 | 1.772 (1.441–2.180) | $y = 2.1645x + 4.4622$ |

[a]Lethal concentration, with 95% confidence intervals in parentheses, causing 50% larval mortality at 120 h after feeding a artificial diet mixed with Bt.

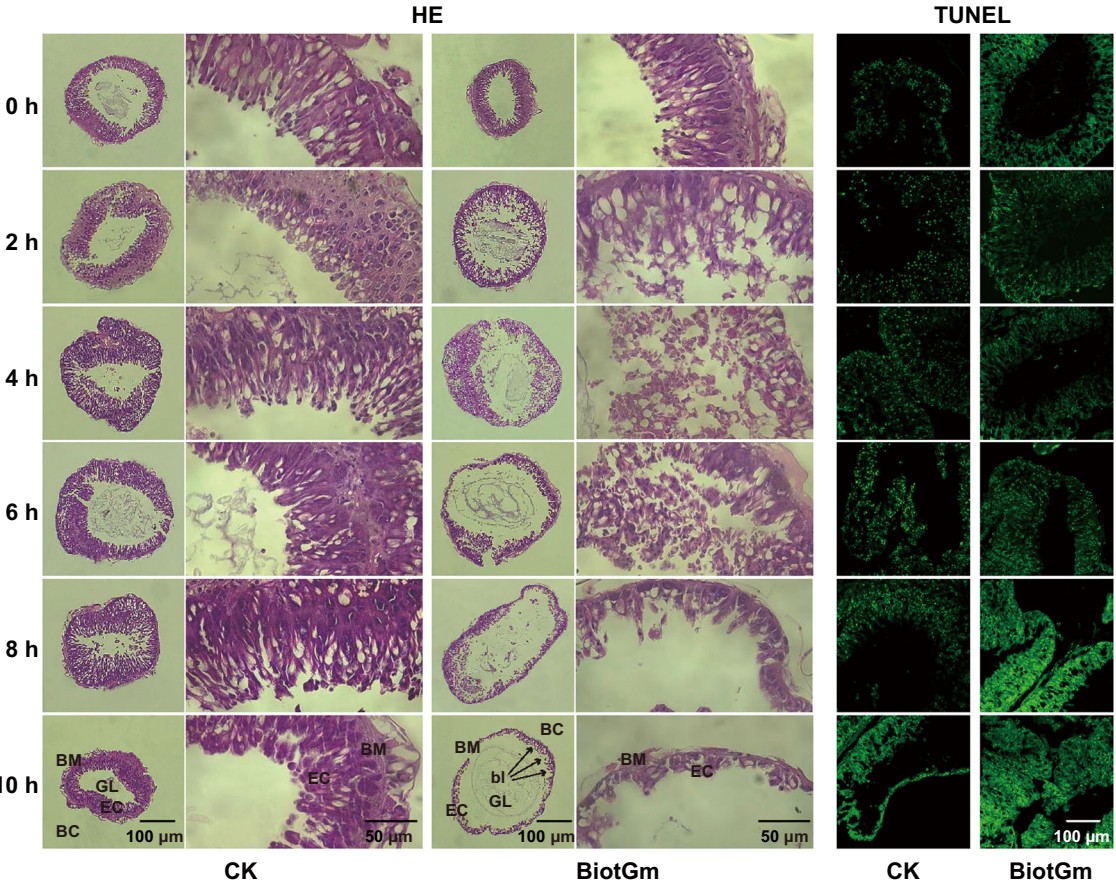

**Fig. 2 | Histopathological changes and apoptosis effects of BiotGm in the GWM larvae midgut.** Whole and local HE and TUNEL (green) staining results of the midgut in the PBS (CK) and BiotGm injection-treated groups at 0, 2, 4, 6, 8, and 10 h. Symbols: GL gut lumen, EC columnar epithelial cells, BM basement membrane, BC body cavity, bl epithelial blebbing. Intensive blebbing or sloughing of the epithelium onto the gut lumen was indicated by black arrows. All HE or TUNEL results were repeated at least twice using independently prepared samples with similar results. The complete DAPI- and TUNEL-stained midgut structure is shown in Supplementary Fig. 3. Scale bar: 100 µm, 50 µm.

observed in the BiotGm-treated group as treatment duration increased. Additionally, the BiotGm-treated group exhibited stronger green fluorescence and apoptosis than the control group at all time points (Fig. 2). The complete DAPI- and TUNEL-stained midgut structures of GWM larvae are shown in Supplementary Fig. 3.

**Activity of Bt against GWM by spraying on and pressing into the comb**

In terms of the spraying application method, of the five tested Bt products, BiotGm always showed the lowest loss rate of combs (<10%) at all tested concentrations, which was significantly lower than that of the control group (all $P < 0.0001$) (Fig. 3a; Supplementary Fig. 4). When the test concentrations of five Bt products were increased to 1000, 2000, and 4000 µg/mL, all loss rates of combs were <5%, which satisfied the needs for GWM control in beekeeping (Supplementary

Figs. 5, 6). Furthermore, to exclude the effect of Bt spore residue in the spraying application method, we filtered out the Bt spores in the process of extracting the crystal protein of BiotGm. The comb loss rate of all five treatment groups with different storage times (<4%) was significantly lower than that of the control groups (nearly 100%) in the 6th month (all $P < 0.0001$) (Fig. 3b). BiotGm demonstrated excellent activity and stability in the above two types of spraying experiments.

For another application method of pressing Bt into a comb foundation, we evaluated the activity of five Bt treated with a series of temperatures (70, 75, 80, and 85 °C) according to the conventional comb foundation preparation method. There was no significant difference in the corrected mortality of GWM larvae between the control (Bt stored at 25 °C) and the heating treatment for 6 h, implying that temperature had not yet affected the insecticidal activity on GWM larvae (Fig. 3c). Based on the above results, 75 °C was adopted to melt

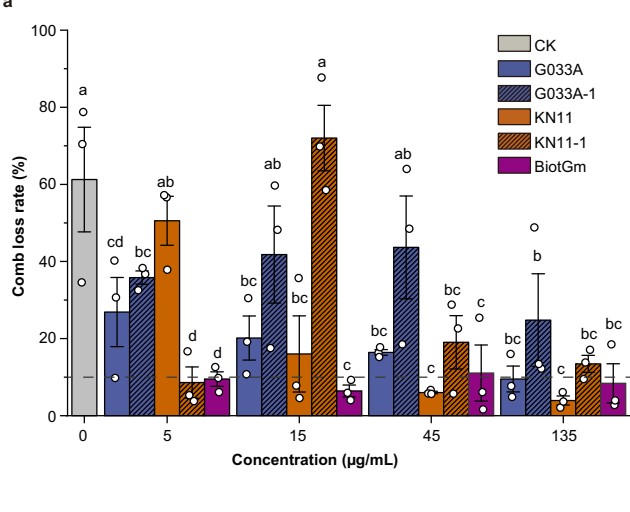

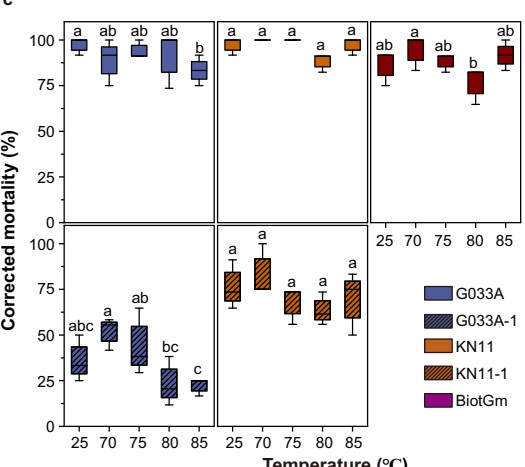

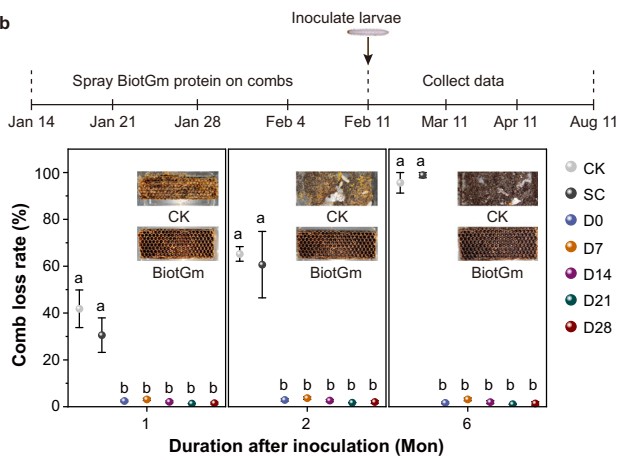

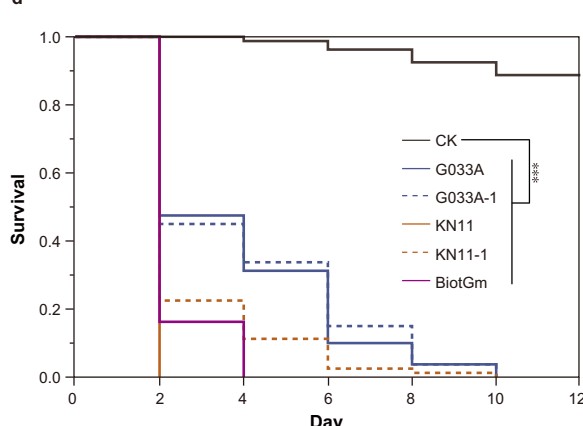

**Fig. 3 | Activity of spraying Bt on comb or pressing Bt into the foundation against the GWM. a** The loss rates of combs sprayed with 5, 15, 45, and 135 μg/mL Bt solution and invaded for 4 weeks by 2nd instar GWM larvae. **b** Comb sprayed with crude BiotGm insecticidal crystal protein and kept at room temperature for 0 (D0), 1 (D7), 2 (D14), 3 (D21) and 4 (D28) weeks followed by invasion by 2nd instar GWM larvae for 1, 2 and 6 months. Combs sprayed with 50 mM Na$_2$CO$_3$ solution as SC and without any treatment were used as CK. Comb loss rate data were replaced by arcsine square-root (sqrt) transformation before performing a two-sided unpaired Student's *t*-test to analyse the differences between groups. **c** The effect of insecticidal activity of five Bt products after continuous heating at 25, 70, 75, 80, and 85 °C for 6 h. Boxplots indicate median (centerline), mean ± SE (box), and the maximum

and minimum values (upper and lower whiskers). Significance was determined by a two-sided chi-square test. Data are presented as the mean ± SE of three independent biological replicates. The same letters above the bars indicate no significant differences between groups ($P > 0.05$). **d** The survival of 2nd instar GWM larvae exposed to the comb foundation containing Bt (4000 μg/g) for 12 days. Comb foundations without any treatment were used as CK. Data were based on four independent biological replicates. Kaplan–Meier survival analysis and two-sided log-rank tests were used to identify differences in survival between groups. All ***$P$-value ≤ 0.001 for CK vs G033A, CK vs G033A-1, CK vs KN11, CK vs KN11-1, and CK vs BiotGm, and exact $P$-values is shown in Supplementary Table 3. Source data are provided as a Source Data file.

beeswax, and five Bt products were separately mixed into it to prepare the comb foundation for testing their insecticidal activity. The survival of 2nd instar GWM larvae exposed to comb foundations containing Bt (4000 μg/g) for 12 days was significantly lower than that of the control group (88.75%) (Fig. 3d; Supplementary Fig. 7; Supplementary Table 3).

### Safety assessment of BiotGm on honey bees

The residues of BiotGm were 0.52–3.86 μg/g in honey, 9.90–22.52 μg/g in pollen and 15.10–20.48 μg/g in comb after spraying 4000 μg/mL BiotGm solution on the comb with honey and pollen (Supplementary Fig. 8). The LC$_{50}$ value of BiotGm for *A. mellifera* larvae was 20598.78 μg/mL, and the LD$_{50}$ value was 617.96 μg/larva (Supplementary Table 4). Both the survival and average weight of *A. mellifera* larvae were not significantly affected by 100 μg/mL BiotGm (>5 × maximum residues) (Fig. 4a, b; Supplementary Table 5). Using the BeeREX calculation, both acute and chronic RQ values were <1 (Supplementary Table 6), indicating that the maximum residue level of BiotGm was safe for *A. mellifera* larvae.

Moreover, consistent with the above results of the risk assessment in larvae, the survival of *A. mellifera* and *A. cerana cerana* worker adults exposed to 100 μg/mL BiotGm for 12 days was not significantly different from that of the control group but was higher than that of the groups treated with 1 or 45 μg/mL dimethoate (Fig. 4c; Supplementary Tables 7, 8). Overall, the general trends of the average daily syrup and pollen consumption of *A. mellifera* and *A. cerana cerana* worker adults after exposure to 100 μg/mL BiotGm were not significantly different compared with those of the control group (Fig. 4d). However, there were certainly some cases in which the average daily syrup and pollen consumption of the 100 μg/mL BiotGm group was significantly lower than that of the control group (Fig. 4d).

### Screening lures for the GWM larvae

Our olfactometer studies revealed that 1st–2nd instar larvae exhibited a stronger preference for the dark comb (comb in which the brood has been reared) and beeswax than pollen and honey (Fig. 5). Specifically,

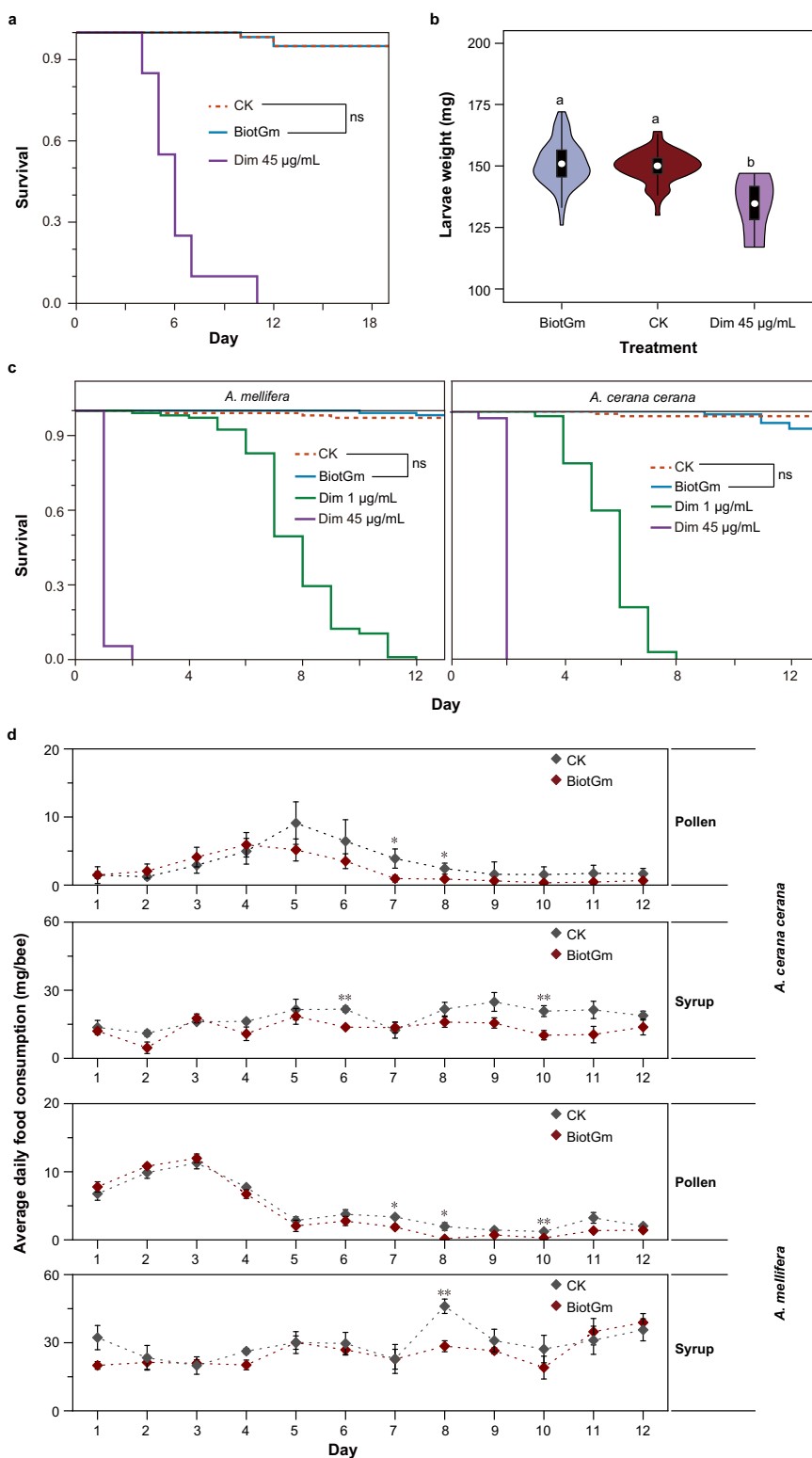

in each test group of dark comb vs. honey, dark comb vs. pollen, and dark comb vs. beeswax, the attraction rate of dark comb to GWM exceeded 90%, 80%, and 60%, respectively; in the test groups of beeswax vs. honey and beeswax vs. pollen, the 1st−2nd instar GWM larvae exhibited a significant preference for beeswax, with attraction rates of 90% and 75%, respectively (all $P < 0.0001$). These results illustrated that the dark comb or beeswax could be used as a lure to attract GWM, and it could be applied as a key component in novel complex attractants to trap and kill GWM.

**Indoor activity of entrapment against GWM**

According to the above attractant results, we developed lure A, mainly consisting of dark comb, and lure B, primarily including beeswax, and then combined them with BiotGm with high insecticidal activity and a trapping device to build an entrapment (Fig. 6a). Then, the entrapment was subjected to a hive simulating the invasion bee colony environment of GWM larvae indoors, and the results showed that it had an excellent attractive effect for GWM larvae and thereby obviously reduced comb invasion (Fig. 6b). Specifically, the survival of larvae in

**Fig. 4 | The safety assessment of BiotGm on honey bee larvae and adults.**
Survival (**a**) and average weight (**b**) of *A. mellifera* worker larvae exposed to 100 µg/mL BiotGm, diets without test substance as control group (CK) and 45 µg/mL dimethoate (Dim). Data were based on four independent biological replicates (each including 12 larvae). The ranges of the violin plots represent the upper and lower quartiles, whereas the points indicate median. The two-sided unpaired Wilcoxon rank-sum test was used to identify the difference in larval average weight between groups. The same letters above the bars indicate no significant differences between groups (*P* > 0.05). **c** Survival of *A. mellifera* and *A. cerana cerana* worker adults after 12 days of exposure to 100 µg/mL BiotGm, syrup without test substance as control group (CK), 1 and 45 µg/mL dimethoate. Data were based on four independent biological replicates (each including 20 adults). **d** Average daily syrup and pollen

consumption of *A. mellifera* and *A. cerana cerana* worker adults after 12 days of exposure to 100 µg/mL BiotGm and the control group (CK). Data are presented as the mean ± SE of five independent biological replicates. Significance of daily syrup and pollen consumption between groups was determined using a two-sided unpaired Student's *t*-test. Exact *P*-values of syrup consumption, *A. mellifera*: **$P_{day8}$ = 0.0036, *A. cerana cerana*: **$P_{day6}$ = 0.0022; *$P_{day10}$ = 0.0104. Exact *P*-values of pollen consumption, *A. mellifera*: *$P_{day7}$ = 0.0146; *$P_{day8}$ = 0.0278; **$P_{day10}$ = 0.0081, *A. cerana cerana*: *$P_{day7}$ = 0.0314; *$P_{day8}$ = 0.0450. All survival data were analysed using Kaplan–Meier survival analysis and the two-sided log-rank test to determine differences, where significance was denoted with a *P*-value > 0.05 (ns). Source data are provided as a Source Data file.

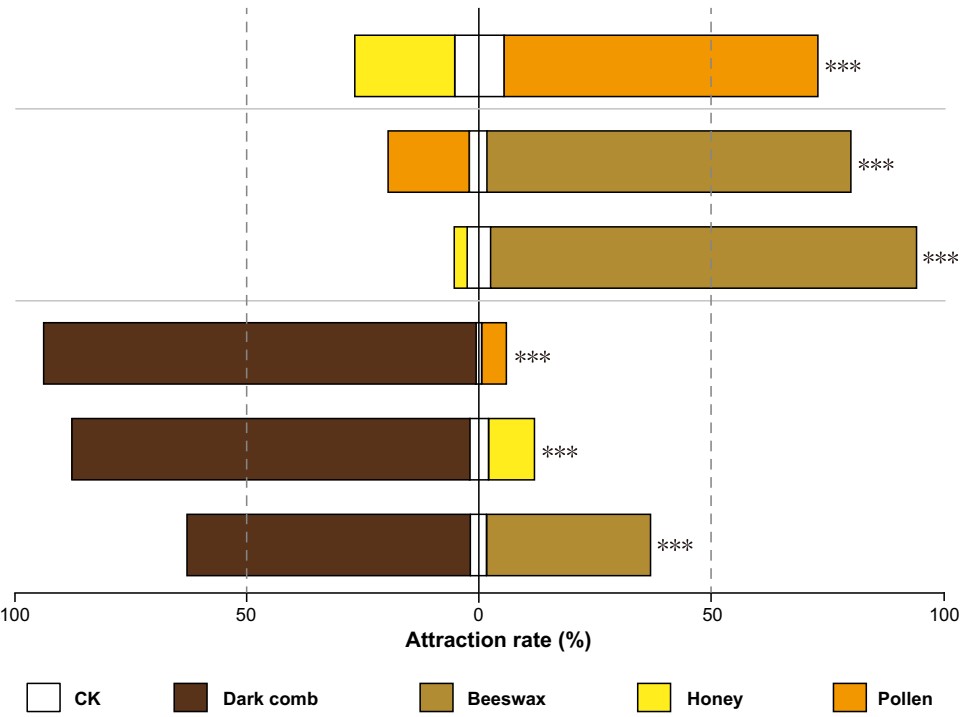

**Fig. 5 | Screening lures for the GWM larvae from four beehive products.**
Attraction rate of GWM larvae to beehive products in a Y-type olfactometer. Data for each test were based on three independent biological replicate experiments (each comprising 48 larvae). Bars represent the percentages of larvae choosing either of the odour sources. A two-sided chi-square test was used to analyse the

significant differences between different beehive products in each test group (***P*-value ≤ 0.001). Exact *P*-values were as following: *P* < 0.0001, (dark comb vs beeswax); *P* < 0.0001, (dark comb vs honey); *P* < 0.0001, (dark comb vs pollen); *P* < 0.0001, (beeswax vs honey); *P* < 0.0001, (beeswax vs pollen); *P* < 0.0001, (honey vs pollen). Source data are provided as a Source Data file.

the lure + BiotGm group was almost 0 after 30 days, which was significantly lower than that in the lure control group and the control group (only comb) (all *P* < 0.0001) (Fig. 6c). The comb loss rate for lure + BiotGm after 120 days was 3.3%, which was significantly lower than that of the lure control (12.2%) and the control (48.5%) ($\chi^2$ = 20.6547, df = 2, *P* < 0.0001) (Fig. 6d).

**Field biocontrol efficacy of entrapment against GWM**

**Initial pilot study.** Before the comprehensive field trial, we selected 14 pilot sites in China for breeding *A. cerana cerana*. The goal was to ascertain the efficacy of the new entrapment method against GWM in beekeeping (refer to Supplementary Figs. 9, 10). After a 5-month observation period, none of the 14 trial sites using the entrapment reported any incidence of GWM, suggesting that the entrapment effectively prevented GWM in beekeeping (see Supplementary Table 9).

**Detailed field trials.** To precisely describe the severity of colony damage caused by the GWM, we selected three sites for systematic research. Before initiating this, we devised a grading system. This

system was designed to classify colony damage into four unique levels based on the harm inflicted by the GWM. A detailed criterion and description of this grading system are available in Supplementary Fig. 11.

These three chosen sites displayed significant disparities in colony populations and the magnitude of GWM damage. This was attributed to variations in habitat conditions, including climate and food sources. The specific results are as follows:

In Nanchang (Jiangxi Province, China), the majority of the test colonies lacked capped brood, and their population size was small (Supplementary Fig. 12). As a result, damage by GWM in the control group was either grade 3 or grade 4 (Fig. 7a).

In Taiyuan (Shanxi Province, China), despite a significant reduction in colony populations in the control group compared to the entrapment group posttrial (Supplementary Fig. 13), no GWM damage was observed in colonies from either group (Fig. 7a). However, 75% of the colonies from the lure control group showed damage from GWM (Fig. 7a).

At the Shijingshan site in Beijing, China, all tested colonies had a substantial number of capped brood and worker bee adults

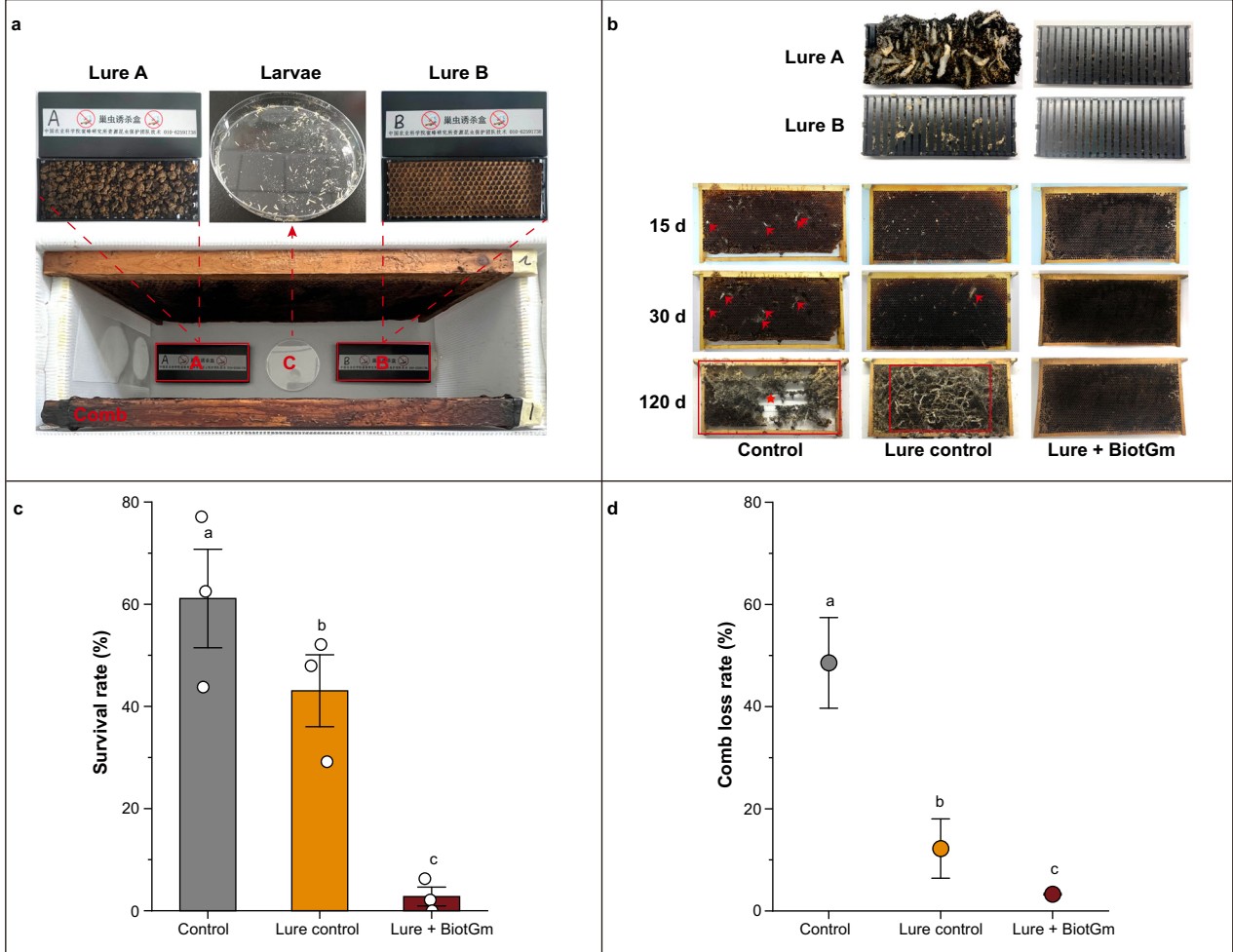

**Fig. 6 | Indoor activity of the entrapment against GWM. a** Setup for trapping device containing lure and schematic of the indoor trapping experiment. **b** Representative images of the attraction and control effects of entrapment after 120 days. The red arrow indicates the silk-lined tunnels produced by the GWM after destroying the comb. The red box shows the area where the combs have been destroyed by the GWM. **c** Survival of GWM larvae after 30 days. **d** The comb loss rate after 120 days employing entrapment. Data are the mean ± SE of three independent biological replicates, with each including 48 larvae and 2 combs. Data are shown as the mean ± SE. A two-sided chi-square test was used to analyse the significant differences in survival rate between different groups. The Kruskal–Wallis test was used to identify the difference in the comb loss rate between groups. The same letters above the bars indicate no significant differences between groups (*P* > 0.05). Source data are provided as a Source Data file.

(Supplementary Fig. 14). The entrapment colonies showed no damage from GWM (Fig. 7a). In contrast, the colonies in both the control and lure control groups suffered from grade 2 damage, where brood combs showed significant signs of bald brood (a condition where GWM larvae tunnel under wax cell cappings leading worker bees to remove the damaged cappings) (Fig. 7a). However, colonies using the entrapment method showed almost no signs of bald brood (Fig. 7a).

Overall, across the three field test sites, using the entrapment method resulted in significant protection against GWM when compared to both the control and lure control groups (Fig. 7a; Supplementary Fig. 15). The effectiveness of the entrapment against GWM in the three trials was over 93% in Shijingshan, Beijing; 100% in Nanchang, Jiangxi; and 100% in Taiyuan, Shanxi. Additionally, we collected data on population (worker bees and broods) and colony weight changes from all test sites before and after the experiment. The findings suggest that colonies treated with lure + BiotGm showed notably positive increases in weight and population compared to the control and lure control groups (Fig. 7b; Supplementary Fig. 16). These results validate the innovative entrapment method's effectiveness in managing GWM pests, confirming its potential in mitigating the detrimental impacts of GWM on beekeeping.

## Discussion

The pervasive presence of GWM, known for their detrimental effects on beekeeping, necessitates accurate global distribution data to guide strategic interventions[15,33–35]. Our research unveiled a larger-than-expected distribution, underscoring the need for effective control measures (Supplementary Fig. 1a). Additionally, we explored the distribution of *A. cerana cerana*, a bee species indigenous to China, revealing its vulnerability to GWM infestation (Supplementary Fig. 1b; Supplementary Table 10). Such a widespread presence accentuates the urgency of deploying effective mitigation strategies to safeguard the economic interests of beekeeping. Our study has been instrumental in addressing some of the key challenges associated with GWM management, particularly in the context of Bt-based interventions.

Although Bt's potential as a biocontrol agent is recognized[20,36–39], applications against GWM remain sparse[15,40,41]. The crucial and first task to successfully apply Bt strains as biological control agents for pest control is to screen Bt strains with high virulence against the target pest. In this study, we evaluated the virulence of five Bt products from three Bt strains against GWM for the first time; among these strains, BiotGm was isolated as a novel strain by our research team for the control of GWM. We found that all three Bt strains showed prominent

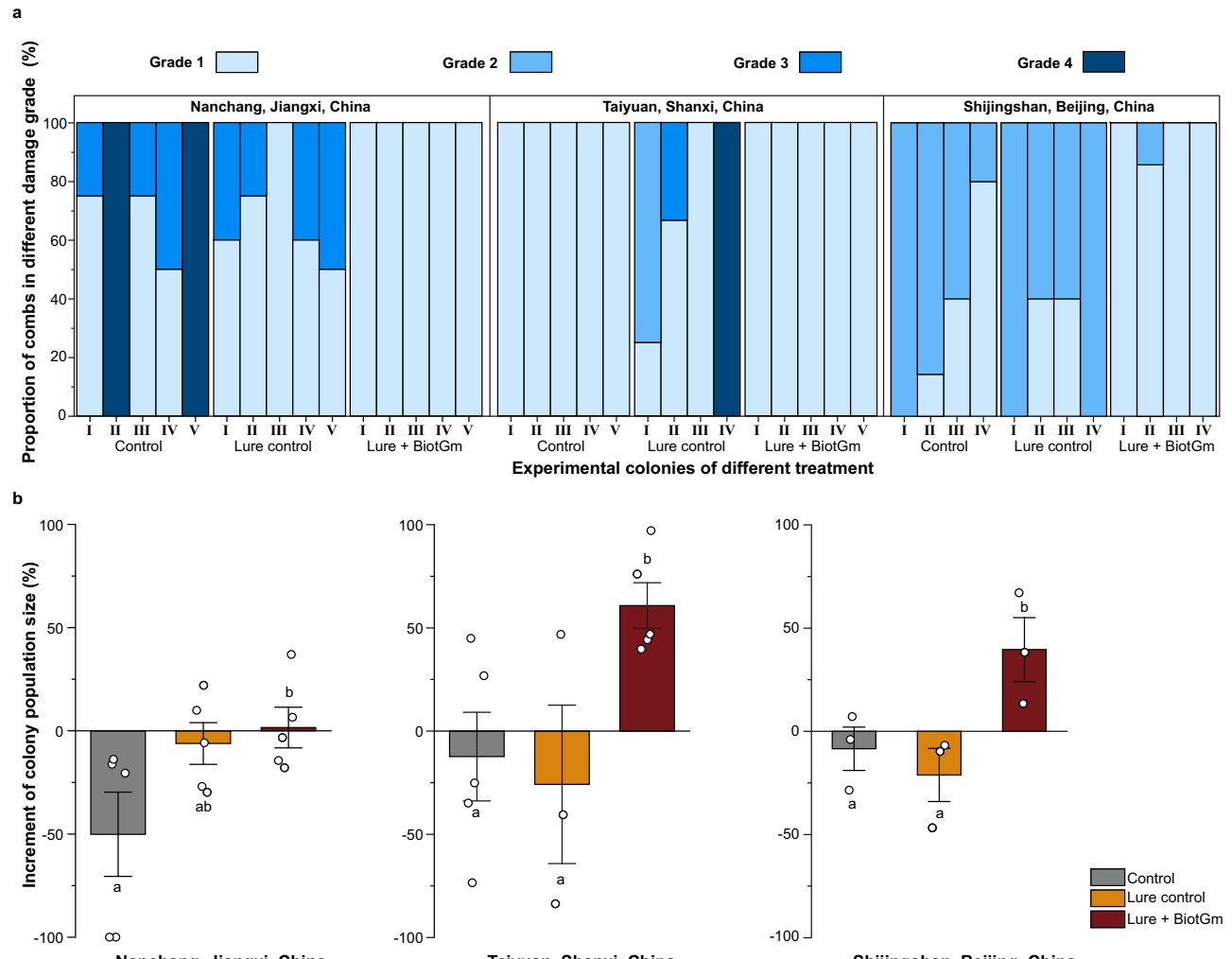

**Fig. 7 | Field biocontrol efficacy of entrapment against GWM. a** Proportion of comb in different GWM damage grades after finishing the trial in different test groups for three field sites. The four severity levels of damage from GWM on colony including: Grade 1−no GWM damage (Supplementary Fig. 11a); Grade 2−severe blad brood occurred in comb by GWM larvae damage (Supplementary Fig. 11b); Grade 3−phenomenon of feeding comb by GWM was obviously found (Supplementary Fig. 11c); Grade 4−colony have been suffered severe damage of GWM and absconding (Supplementary Fig. 11d). **b** Changes of population size before and after the trial in different test groups from three field sites. Each field test site include five independent colony replicates for all treatments ($n = 45$ colonies), and data are the mean ± SE. A two-sided Mann–Whitney U test was used to analyse the significant differences between different groups of colony population size increment. Same letters above bars indicate no significant differences between groups ($P > 0.05$). The distribution of 2950 colonies of *A. cerana cerana* from the 14 test sites in China for field trails shown in Supplementary Fig. 9 and Supplementary Fig. 10. Source data are provided as a Source Data file.

insecticidal activity against 2nd instar GWM larvae at 120 h post-treatment. Among them, our novel strain, BiotGm, showed exceptional efficacy (Table 1). Moreover, their control efficacy was separately further tested on combs using a conventional spray method. Consistent with our above laboratory results, they all obtained good wax moth control over a 4-week assay period, and BiotGm showed the highest insecticidal efficacy (Fig. 3a).

A crucial challenge with Bt applications is the lack of a standardized bioassay technique to measure its efficacy against the GWM[42–45]. In this investigation, we innovatively addressed this issue by introducing a novel, reproducible bioassay method. The technique entails the creation of a modified synthetic diet and a survival environment that are conducive to the growth and development of the larvae in vitro and is based on their nutritional needs and rearing conditions. Compared to the previous bioassay for multiple GWM larvae exposed simultaneously to Bt with a comb or common diet as the carrier[43,46,47], the modified synthetic diet and the rearing method in vitro are more precise in measuring the toxicity of Bt, as they eliminate various confounding factors, such as interindividual competition and escape

behaviour, and the survival of the negative control group reached up to 95%; however, the obtained data could provide more effective and targeted measures to manage wax moth infestations (Fig. 1; Table 1; Supplementary Fig. 17). Altogether, our newly established approach can be employed in future investigations to assess the effectiveness of Bt or other agents against GWM. Additionally, this new methodology has applications beyond just Bt efficacy measurement; it presents a platform to delve deeper into GWM larvae nutritional dynamics, opening avenues for future pest management strategies.

In the development of novel Bt strains, it is critical to ensure that their utilization does not inflict any unintended harm on nontarget organisms[48–51]. To this end, we conducted extensive toxicity studies in the laboratory, following standardized protocols, to assess the potential risks of BiotGm, a new Bt strain, to honey bees, which are a crucial nontarget species. During our work, we chose *A. mellifera* and *A. cerana cerana*, the most common honey bee species in China's agroecosystems, as nontarget organisms and subjected them to extreme concentrations of BiotGm. We monitored their survival, behaviour, and reproductive capacity closely, and the results

unequivocally demonstrate that BiotGm has minimal effects on honey bees while exhibiting potent toxicity against GWM (Fig. 4). These findings offer crucial support for BiotGm as a viable candidate for controlling GWM. However, additional research, including field evaluations, is essential in the future.

Efforts to optimize Bt toxin for pest control have led to enhanced formulations and novel application methods[27]. One area of focus is developing more effective and targeted application methods that minimize the risk of exposure to nontarget organisms while maximizing the control of pest populations. For example, encapsulation of Bt spores in a polymer matrix can protect the spores from environmental degradation and improve their efficacy. The use of bionanotechnology has also been explored to create targeted delivery systems that can enhance the specificity and efficacy of Bt[26]. In our research, we devised an alkaline lysis-filtration technique to extract a pure BiotGm insecticidal protein, discarding nutrients and spores. Spraying this solution onto comb, we confirmed its persistent insecticidal properties over 4 weeks (see Fig. 3b). By omitting spore residues, we not only preserved its efficacy but also negated risks to beneficial organisms such as bees. Such innovations, suggesting improved safety and effectiveness of Bt protein without spores, pave the way for a more sustainable and eco-friendly application of Bt toxin in pest management.

Recent research has focused on the use of attractants to boost Bt efficacy in pest management[52]. The integration of Bt with specific pest lures not only enhances pest control efficacy but also offers several benefits: (i) improved pest contact with the Bt formulation, (ii) heightened precision and efficiency of Bt applications, and (iii) reduced nontarget organism exposure to the biopesticide. However, studies integrating attractants with Bt to control GWM remain limited. In our work, we employed a specialized olfactometer, identifying a substance significantly attracting GWM (Fig. 5). We then devised a biocontrol entrapment strategy combining lures, Bt, and a protective device shielding bees from the attractants. Lab tests supported that this entrapment is highly effective against GWM invasions (Fig. 6). However, real-world field conditions, influenced by factors such as climate and colony health, can impact its success. Thus, field testing remains imperative to validate its efficacy in natural settings.

In recent advances in sustainable pest control, the combined application of Bt with specific attractants tailored for target pests has emerged as a groundbreaking strategy[52–54]. This dual approach not only amplifies the chances of pests encountering Bt, enhancing its pest-killing efficacy but also refines the Bt formulations, ensuring that they are more precise in their action. Moreover, this methodology significantly reduces unintended exposure to beneficial organisms, underpinning its commitment to environmentally conscious pest management. Building on the strength of these benefits, our research identified a unique compound that GWM have a pronounced attraction to (Fig. 5). Leveraging this discovery, we have developed an all-encompassing biocontrol strategy for GWM, thoughtfully integrating these lures with Bt and introducing a protective mechanism to prevent bees from inadvertent exposure (Supplementary Fig. 18, 19). Both lab and field trials stand testament to the effectiveness of this innovative entrapment technique against GWM invasions, with supporting data detailed in Fig. 6 and Fig. 7. As our understanding deepens, this integrated approach illuminates the path forwards, underscoring the potential of Bt in orchestrating a new chapter in sustainable GWM management.

In conclusion, our comprehensive study elucidates the promising potential of a novel Bt strain in combating GWM. Exhibiting high toxicity specifically towards the target pest yet safeguarding honey bees, this Bt strain presents a dual benefit. Additionally, its properties facilitate hassle-free storage and transport, making it an optimal choice for beekeepers. Various application methods, such as spraying, integration into the comb foundation, and bait mechanisms, make this

strain a versatile tool for both in situ bee colony protection and preventive measures during the ambient storage of combs and honey. However, like every scientific endeavour, our study is not without its shortcomings. A salient limitation is our current lack of clarity regarding the specific substance within the device's lure core that exhibits attractant properties. Moving forwards, it is imperative for our research to delve deeper into understanding this mechanism. By unravelling the intricacies of this alluring component, we aspire to amplify the device's efficacy, making it an even more potent tool in the war against GWM infestations.

## Methods

### *Bacillus thuringiensis* and insects

Bt G033 A, G033A-1, KN11, KN11-1 and BiotGm were provided by the Institute of Plant Protection, Chinese Academy of Agricultural Sciences (Beijing, China). G033A is an engineered Bt strain that exhibits strong insecticidal activity against both lepidopteran (*Spodoptera exigua*, *Plutella xylostella* and *Helicoverpa amigera*) and coleopteran (*Pyrrhalta aenescens*) pest larvae, and it was developed by introducing the recombinant plasmid pSTK-3A containing the *cry3Aa7* gene encoding a coleopteran-specific insecticidal protein into wild *B. thuringiensis* subsp. *aizawai* G03, which contains the c*ry1Aa*, *cry1Ac* and *cry2Ab* genes and is highly toxic to lepidopteran insect pests[55]. KN11 belongs to *B. thuringiensis* subsp. *azawai* with high insecticidal activity against a diverse range of lepidopteran pest larvae (*Hyphantria cunea*, *Spodoptera exigua*, *Spodoptera litura*, *Chilo suppressalis*, *Spodoptera frugiperda*, *Plutella xylostella*, *Helicoverpa armigera*, etc.)[56]. BiotGm is a novel Bt strain with enormous potential applications in pest management and was isolated by our research team in China. The five Bt products are wettable or primary powders, among which G033A (primary powder), KN11 (primary powder) and BiotGm (wettable powder) are high purity without any treatment, while G033A-1 and KN11-1 are commercial formulations (wettable powders) of their primary powders that have been processed and marketed. The detailed information of the five Bt is listed in Supplementary Fig. 20 and Supplementary Table 11.

The GWM larvae were obtained from the Institute of Apicultural Research, Chinese Academy of Agricultural Sciences (Beijing, China) (Fig. 1). The GWM larvae were reared on an artificial diet and maintained in darkness in an incubator at $30 \pm 1\,°C$ and $60 \pm 5\%$ relative humidity (RH). The composition of the artificial diet was based on the formula previously described[57].

The worker bee adults and larvae of *A. mellifera* were obtained from an apiary (N39°59′35.33″, E116°11′59.74″) at the Institute of Apicultural Research, Chinese Academy of Agricultural Sciences, Beijing, during June-August 2021, and *A. cerana cerana* worker adults were obtained from an apiary (N40°13′35.10″, E116°04′15.63″) in Shijingshan district, Beijing, during August-September 2021. All test honey bees were collected from healthy colonies that were managed per common best management practices for the region (including feeding when necessary, managing diseases/pests, etc.). The method and timeline of *A. mellifera* worker larvae were reared in vitro following the same procedure described previously, and the queens that were caged on a wax comb for 24 h to lay eggs were sisters obtained from the same lines[58]. The worker adults of *A. mellifera* and *A. cerana cerana* used for the test were all newly emerged and were reared in wooden cages $(9 \times 9 \times 10\,cm)$ with mesh on two sides under in vitro conditions according to previously modified methods from the standardized test guidelines OECD 245[59].

### Bioassay of Bt against GWM larvae

The optimal bioassay diet was based on a conventional artificial diet of GWM larvae and the artificial diet formulation of *Spodoptera frugiperda*[60,61]. The bioassay diet consisted of 10 g yeast extract powder, 25 g milk powder, 50 g wheat flour, 50 g corn powder, 50 g wheat

bran, 2 g sorbic acid, 26 g casein, 21 g agar powder, 25 g honey, 25 g beeswax, 30 g glycerol, 900 mL ddH$_2$O, 1.5 mL formaldehyde, 3 mL acetic acid, 60 mL ascorbic acid, and 0.5 mL 40 g/L thiabendazole.

The preferred bioassay method was constructed referring to the methods of *S. frugiperda* and *H. armigera*[62,63] (Fig. 1). Five Bt products were dissolved in ddH$_2$O and diluted to suspensions of various concentration gradients. Then, 2 mL of the above diluted suspension was added to the 10 g bioassay diet and stirred evenly. Based on the preliminary experiments the following concentrations were designed to determine the LC$_{50}$: GO33A: 0.5, 1, 2, 4, and 8 μg/g; GO33A-1: 2, 4, 8, 16, and 32 μg/g; KN11: 0.5, 1, 2, 4, and 8 μg/g; KN11-1: 1, 2, 4, 8, and 16 μg/g; and BiotGm: 0.5, 1, 2, 4, and 8 μg/g. The mixed diet was stored at room temperature for 3 h to evaporate the excess water. The bioassay diet was evenly divided into a 12-well sterile cell culture plate and pressed to one side of the bottom of the hole using a spoon. The robust 2nd instar larvae were inoculated into the wells with one larva per well. The plates were placed in an incubator at 30 ± 1 °C, 60 ± 5% RH, and darkness. In addition, ddH$_2$O (2 mL) was added to the 10 g diet as a control. There were 4 replicates per treatment and 12 larvae per replicate. The number of dead and living larvae was checked daily by visual inspection. The dead larvae infected with Bt are black. The mortality data for 120 h were corrected using the natural mortality in the control via Abbott's formula[64]. The LC$_{50}$ value for 120 h was calculated using a previously described procedure[65].

### Preparation of the genomic DNA, sequencing, and computational analysis for BiotGm

Genomic DNA from BiotGm was prepared using the procedure reported[66]. The BiotGm genome was sequenced using the Pacific Biosciences RS II (Pacific Biosciences, Menlo Park, CA, USA) sequencing platform. The sequencing depth was ~200-fold coverages. De novo assembly of PacBio reads was carried out using RS_HGA-P_Assembly.3 protocol included in SMRT portal. Genome annotation was performed by the NCBI Prokaryotic Genome Automatic Annotation Pipeline. Composition vector tree (CVTree) phylogenetic analysis was performed using a free, web-based tool-the *Bacillus* Typing Bioinformatics Database (https://btbidb.com)[67]. The genome data of other *B. thuringiensis* strains were selected in the database to generate the phylogenetic tree.

### Histopathological analysis and apoptosis in the midgut of GWM larvae exposed to BiotGm

Histopathology assays were carried out according to a previous description[68]. The 4th instar GWM larvae were injected with 10 μL BiotGm (100 μg/mL) or PBS as the treatment group and control group, respectively. The midguts of larvae at 0, 2, 4, 6, 8, and 10 h after injection of BiotGm or PBS were dissected under bright field optics using a sterilized blade. The midgut tissues were suspended in ice-cold 4% PFA (Sigma–Aldrich, USA) overnight at 4 °C. After fixation, the midgut tissue samples were dehydrated through an ethanol series, embedded in paraffin wax and cut into 6 μm sections, and the sections were stained with haematoxylin and eosin (Sigma–Aldrich, USA). Images were captured with a ProgRes 3012 digital camera on a Leica Diaplan microscope.

The midgut tissue samples were prepared as described above. Apoptosis was conducted using a modified method according to a previous method[69]. The midgut tissue samples were soaked in 4% PFA (Sigma–Aldrich, USA) for 1 h, transferred to 30% sucrose in PBS, and stored overnight at 4 °C. After dehydration, the midgut tissue samples were embedded using an embedding agent (200 mL, Leica, Germany) and cut into 6 μm frozen sections at −20 °C using a Leica microtome (Leica CM1900, Leica, Germany). TUNEL assays were carried out using the In Situ Cell Death Detection Kit (Roche, Indianapolis, IN, USA). The TUNEL reaction mixture was freshly prepared and applied to sections and then incubated for 2 h at 37 °C in the dark in a humidified chamber.

Finally, DAPI (Sigma–Aldrich, USA) was used for counterstaining for 6 min to observe the distribution of cells. All prepared slides were examined under a laser confocal microscope (Leica SP8, Leica, Germany) and scanned after immunofluorescence staining. Midgut cells were imaged at 20 × magnification as partial and whole structures.

### Indoor activity trial of spraying Bt and crystal protein on comb against GWM

Five Bt products were dissolved in ddH$_2$O and diluted into suspension at 5, 15, 45, and 135 μg/mL concentrations. These set values are 2.8-fold, 8.5-fold, 25.4-fold and 76.2-fold of the LC$_{50}$ of BiotGm against GWM larvae, respectively. The 2.5 mL Bt suspension of different concentrations was sprayed on the comb (12 × 4 × 2 cm), which was placed at room temperature overnight to air dry the excess water and put into the plastic container (14.5 × 9 × 6.5 cm) with thirty 2nd instar larvae. In addition, ddH$_2$O was sprayed as a control, and each group was performed in 3 replicates. The plastic container was placed in a climate box at 30 ± 1 °C, 60 ± 5% RH, and darkness. The loss rate of combs was measured at the 4th week after larval inoculation, and the damage to the comb was recorded by photographing. To determine the concentration used in the field colonies and further reduce the loss rate of the comb, we increased the Bt concentration to 1000, 2000, and 4000 μg/mL. The loss rate of combs was measured at the 4th, 5th, 6th, 7th and 8th weeks after larval inoculation. The formula $V = \frac{Wo - Wn}{Wo - P} \times 100\%$ was used to calculate the percentage of comb loss, where $V$ is the loss rate, Wo is the initial weight of the comb and plastic container, Wn is the weight of the comb and plastic container after being treated for different times, and P is the weight of the plastic container.

The crystal protein preparation was carried out as previously described with modification[19]. Briefly, the washed spore-crystal mixture of BiotGm was solubilized by incubation in ten volumes of buffer containing 50 mM Na$_2$CO$_3$, 10 mM EDTA, and 10 mM DTT (pH 10.0) at 37 °C for 30 min. The protein solution was centrifuged at 13,680 ×g for 10 min, and then the supernatant was adjusted to pH 5.0 and incubated at 4 °C for 1 h. Next, the protein pellet was collected by centrifugation at 13,680 ×g for 10 min and further washed three times with sterile ddH$_2$O. Finally, the precipitate was dissolved in 50 mM Na$_2$CO$_3$ buffer (pH 10) and filtered through a 0.22 μm membrane to remove the vegetative cells and spores. The above crystal protein was prepared before use and analysed using 10% SDS–PAGE, and the concentration of crystal protein was quantified using a standard bicinchoninic acid assay kit (Thermo Scientific, USA). To evaluate the indoor control effect of insecticidal crystal protein on the GWM, the spraying method was applied according to the above description. A total of 2.5 mL of 9 mg/mL crystal protein solution was sprayed on the combs, which were then kept at room temperature for 0, 1, 2, 3 and 4 weeks. The combs of the control groups were treated without any processing (CK) or with 50 mM Na$_2$CO$_3$ buffer (SC). Second instar larvae were inoculated into the treated combs simultaneously. The loss rate of combs was measured at 1, 2, and 6 months after larval inoculation. Each group had 4 replicates, and each replicate contained 12 larvae.

### Indoor activity trial of pressing Bt into the comb foundation against GWM

The five Bt products were treated at 70, 75, 80, and 85 °C for 6 h and then used to test their insecticidal activity to evaluate their heat stability. Bt was stored at 25 °C as a control, and the test concentration of all groups was 20 μg/g. The bioassay of the insecticidal activity followed the same procedure as described above.

Bt (0.6 g) was mixed with 150 g of beeswax and kept at 70 ± 5 °C, and then the melted beeswax containing the Bt was poured into the hand-pressed comb foundation machine to press the comb foundation, and the final concentration of Bt was 4000 μg/g. The comb foundations were cut into rectangular blocks (12 × 6 cm) and put into a

plastic container with twelve 2nd instar larvae. The comb foundation without Bt was used as the control, 4 replicates were conducted for each group, and the survival rate of larvae was calculated every 48 h.

## The safety of BiotGm on honey bees

The spray-treated combs using 30 mL 4000 μg/mL BiotGm suspension were placed at room temperature to evaporation the excess water. Then, honey, pollen and comb samples were collected to determine the BiotGm residue using the method previously described with modification[70]. 1 g of the sample was homogenized in 9 mL sterile saline. This suspension was used for serial dilution, and the 100 μL of each dilution (n = 3) was plated on LB agar plates and incubated 24 h at 37 °C and then phenotypic Bt colonies were counted. Each sample was set with 3 comb repeats. To confirm the accuracy of the results of BiotGm residue, these colonies were randomly selected and identified by amplification and sequencing the fragments of 16 S rRNA using primers of 27 F and 1492 R (27 F: AGAGTTTGATCCTGGCTCAG; 1492 R: TACGGCTACCTTGTTACGACTT). These obtained 16 S rRNA sequences were shown in Supplementary Table 12 and analyzed by the NCBI BLAST online tool (https://blast.ncbi.nlm.nih.gov/Blast.cgi).

The acute and chronic toxicity trials of BiotGm on *A. mellifera* larvae were performed according to the method described in detail in our previous study[71]. Three in vitro larval diet compositions (A, B, C) were used in the test and administered on different days (D). Diet A (D1–D2): royal jelly 44.25%, glucose 5.3%, fructose 5.3%, yeast extract 0.9% and water 44.25%; diet B (D3): royal jelly 42.95%, glucose 6.4%, fructose 6.4%, yeast extract 1.3% and water 42.95%; diet C (D4–D6): royal jelly 50%, glucose 9%, fructose 9%, yeast extract 2% and water 30%. On D1, the 1-day-old larvae were transported from the comb to sterile tissue culture plates (STCPs) with 20 μL of diet A in each cell well. On D2, the larvae were not fed an additional diet. On D3, 12 robust larvae were selected per replicate of each treatment and used for the acute and chronic toxicity trials. For the acute toxicity trial, on D4, the 4 day-old larvae were fed 30 μL of diet containing the corresponding concentration of BiotGm for the treatment, while D3, D5 and D6 larvae were fed normal diets (no test solution) of 20 μL, 40 μL and 50 μL, respectively, and mortalities were checked and recorded at the time of feeding on D5, D6 and D7. For chronic toxicity trials, over the next 4 days (D3-D6), the larvae were fed 20 μL (D3), 30 μL (D4), 40 μL (D5), and 50 μL (D6) of diet containing the corresponding concentration BiotGm for the treatment, and mortalities were checked and recorded at D4-D18.

The following concentrations were designed to determine the LC$_{50}$: 5000, 10000, 20000, 40000, and 80000 μg/mL. The correction of mortality data for 72 h and the calculation of LC$_{50}$ for 72 h were based on the above methods. Based on BiotGm maximum residue values, the following treatments were conducted for each test solution to determine the chronic toxicity of BiotGm on *A. mellifera* larvae: 100 μg/mL BiotGm, a control group, and a positive control group (45 μg/mL dimethoate). Five hive replicates with 12 robust larvae per source hive were conducted for each treatment. The comparisons of larval weight were only performed for individuals who survived to day 7.

The risk quotients (RQs) for BiotGm on *A. mellifera* larvae followed the same procedure of test compounds on *A. mellifera* larvae risk as described previously[71]. Based on the larval LD$_{50}$, the no observed adverse effect concentration (NOAEC) and the maximum residue values in honey and pollen using the US Environmental Protection Agency's BeeREX model (https://www.epa.gov/sites/production/files/2015-11/beerexv1.0.xlsx) were used to calculate the RQs.

Based on the maximum residue values of BiotGm, a 100 μg/mL concentration solution was designed to determine the chronic toxicity of BiotGm for *A. mellifera* and *A. cerana cerana* worker adults. The emerging worker bee adults were reared in cages and provided two feeders per cage, containing 2 mL of syrup (50% w/v sucrose solution) and the other containing 2 g of pollen paste. Syrup with 100 μg/mL BiotGm was the test group, syrup without any test solution was the control group, and syrup with dimethoate 1 or 45 μg/mL was the positive control. Twenty adults were tested per cage, and five replicates (cages) were established for each treatment. The adults were fed the syrup containing different test solutions for 12 days, and the syrup and pollen paste were replaced daily. After taking the feeder out of the cage at noon the next day, it was weighed. In addition, evaporation control was established to account for water loss when determining syrup and pollen diet consumption.

## Attractiveness of different beehive products for GWM larvae

In the Y-type olfactometer (transparent, colourless, and odourless polyethene), six choice tests were performed with the 1st–2nd instar GWM larvae, including (i) dark comb vs. beeswax; (ii) dark comb vs. honey; (iii) dark comb vs. pollen; (iv) beeswax vs. honey; (v) beeswax vs. pollen; and (vi) pollen vs. honey. The behavioural assays were performed as described in a previous study with some modifications[72] (Supplementary Fig. 21). In brief, two beehive products were randomly placed in the odour source containers of two randomly selected arms for each test, and the odour source container of an arm without any test substance was used as a control. The basic structure of this Y-type olfactometer consisted of a central circular opening (5 cm diameter) with three arms (6 cm long, 2 cm height, and 1.5 cm diameter) posed at a 120° angle. To start a measurement, forty-eight robust 1st–2nd instar GWM larvae were placed at the base of the Y-type olfactometer central circular opening, and then the Y-type olfactometer was placed in a climate box at 30 ± 1°C, 60 ± 5% RH, and darkness. After 30 min, the number of larvae per odour source container was recorded to calculate the attraction rate. In the six choice tests, all beehive products were weighed to 0.4 g, and 3 replicate assays were performed per test. The Y-type olfactometer was cleaned using 70% ethanol and oven-dried before each choice test session. The formula $A = \frac{B}{I} \times 100\%$ was used to calculate the attraction rate, where $A$ is the attraction rate, I is the total number of tested larvae, and B is the number of larvae per odour source container. Larvae that did not make a choice were excluded from the calculation.

## Biocontrol efficacy of an entrapment containing BiotGm against GWM

**Design and production of trapping devices.** According to the habit of the 1st instar larvae of the greater wax moth feed on wax crumbs at the hive bottom, we designed a trapping device to store the lure and Bt. The design and production process of the trapping device was as follows: (i) measure the distance from the hive bottom to the lower frame beam of the comb, considering conditions under which honey bees could not enter and leave freely, but the greater wax moth larvae could; (ii) use the 3Ds Max 2018 software (Autodesk, Inc., San Rafael, CA, USA) to design draft, and then use 3D printer-CREALITY CR-200B (Shenzhen Chuangxiang 3D Technology Co., Ltd., Shenzhen, China) and ABS (CREALITY 3D., Shenzhen, China) material for physical device printing; (iii) carry out the injection moulding production of the trapping device (Beijing Huazheng Longtai Technology Co., Ltd., Beijing, China). The injection moulding material was black high rigidity pp. The parameters of the trapping device are shown in Supplementary Fig. 18.

**Indoor trapping experiments.** According to the results from our attractiveness behavioural assays, two lures (lure A and lure B) containing BiotGm were developed. (i) Lure A + BiotGm: 300 mL BiotGm solution with a concentration of 200 μg/mL was sprayed on 600 g lure A (400 g dark comb powder and 200 g pollen), which was placed overnight at room temperature to evaporate the excess water so that the final concentration of BiotGm in lure A was 100 μg/g. (ii) Lure B

+BiotGm: 0.01725 g BiotGm powder was mixed with lure B (150 g melted beeswax, 9 g comb powder, 3 g pollen powder, 3 g milk powder, and 7.5 g honey), which was poured into a hand-pressed comb foundation machine to press the comb foundation. The resulting comb foundation was cut into tablets (12 × 5 × 0.2 cm), which were labelled attractant tablets with 100 μg/g BiotGm. Two lures containing 100 μg/g BiotGm were placed in two trapping devices to complete the preparation of entrapment A (trapping device containing lure A + BiotGm) and entrapment B (trapping device containing lure B + BiotGm). For the treatment group, entrapments A and B were placed on each side of the bottom of the pollination box (Fig. 6a). The trapping device containing lure A or lure B without BiotGm, as a lure control, was placed at the same position. The pollination boxes without any treatments were used as control groups. The combs (without GWM) were weighed and placed in each pollination box of each group. A culture dish (90 mm in diameter without a lid) with forty-eight robust 1st–2nd instar GWM larvae was placed at the centre of the bottom of the pollination box (Fig. 6a). The distance from the culture dish to entrapment A, entrapment B and the comb was roughly equidistant. The pollination boxes of all groups were placed in a climate box at 30 ± 1 °C, 60 ± 5% RH, and darkness. After 30 days, the number of GWM larvae surviving in the pollination box of all groups was checked to calculate the survival rate of each group. To obtain more significant and accurate results for the comb loss rate, we extended the test time to 120 days to weigh each comb in each pollination box of each group and calculated the comb loss rate. Three replicates were conducted for each group.

**Initial field trapping experiments.** A preliminary field test was conducted to determine the expected efficacy of the entrapment against GWM. Fourteen field test sites were selected in the main breeding areas of *A. cerana cerana* in China (Supplementary Fig. 9, 10 and Supplementary Table 9). The materials and accompanying instruction manual of the entrapment were mailed to the beekeepers in May 2022 (before the peak of GWM occurrence) for the trial. All efficacy feedback information from the 14 test sites was collected from November to December 2022. The obtained information was collated and analysed to determine the field efficacy of the entrapment.

**Formally field trapping experiments.** The entrapment preparation and experimental treatments were conducted in accordance with the methodology employed during the indoor trapping experiments based on the obtained results. These treatments included a control group, a lure control group, and a lure + BiotGm treatment group. Nevertheless, importantly, the trial in question was carried out in a real-world setting, specifically within field colonies. To ensure the validity and generalizability of our findings, we carefully selected three *Apis cerana cerana* apiaries that are representative of different regions of the main breeding areas of *A. cerana cerana* in China, including one apiary situated in southern China (Nanchang, Jiangxi, 28°33'28.5"N, 115°56'0.55"E) and two apiaries located in northern China (Shijingshan, Beijing, 39°57'59.57"N, 116°8'27.1"E and and Taiyuan, Shanxi, 37°47'29.89"N, 112°36'38.95"E) (Supplementary Fig. 22). The field experiment was carried out from July 2023 to September 2023. To ensure the utmost fidelity to natural field conditions, the colonies at the three field trial sites were untouched by human observation and treatment prior to the commencement of the test. In each field trial site, a total of 15 colonies, unaffected by GWM infestation, were chosen at random to serve as the test colonies. Subsequently, each treatment group was randomly assigned five colonies for the purpose of repetition and labelling. Following the completion of the necessary preparations for the test colonies, an evaluation was performed to determine the strength of each test colony. The colony strength and weight were recorded in July, i.e., 1 day before the beginning of the field trial (T0), and in September after 1 month of testing (T1) according to

the standard method in the BEEBOOK[73]. The hives were opened at dawn, when worker bees were not in flight, and the colony and per comb in frame were photographed at each side. The pictures were later subjected to analysis using ImageJ software (ImageJ, Maryland, USA), employing the previously established methodology[74]. The surface of adult bees and capped broods was considered for the strength evaluation. Trapping devices containing lure + BiotGm were positioned at the bottom of the hives corresponding to the test colony from the entrapment treatment. The trapping device containing lure without BiotGm, as a lure control, was placed at the same position. The hive of the control group did not have any device. To avoid the influence of climate change on the occurrence and severity of GWM pest infestation and to enhance the control effectiveness of our entrapment in practical field settings, a culture dish (90 mm in diameter without a lid) with sixty robust 1st–2nd instar GWM larvae was placed into the bottom of the hives for each colony within each experimental group according to the foraging behaviour of GWM, and the positioning method employed for the culture dish within the hive adhered to the methodology utilized in the preceding test conducted indoors. The experiment was carried out for a duration of 1 month, during which no interventions or treatments were administered to the colonies. To assess the field efficacy of the entrapment against the GWM, the severity of GWM infestation on colonies was categorized into four grades based on the standard methods for wax moth research in the BEEBOOK[16]. These four damage grades included the following: (i) grade 1, no GWM damage was found in colonies; (ii) grade 2, severe symptoms of bald brood (GWM larvae tunnel under wax cell cappings, causing worker bees to remove the damaged cappings) on the comb caused by GWM were identified, but the lack of observable traces indicated that GWM fed on comb and honey bee products; (iii) grade 3, traces (GWM larvae feed on combs, cast larval skins, pollen, and some honey) of GWM feeding on the comb and honey bee products were observed, although they did not result in colony absconding; and (iv) grade 4, the colony suffered severe damage from GWM and bees fled from the hive. The representative values of the specific grade for the four grades were 0, 1, 2, and 3. After a period of 1 month, the level of damage inflicted by the GWM on the comb per test colony from three field sites was assessed utilizing the established classification standard, and the damage grade of the GWM and strength and weight for each test colony were thoroughly recorded for every treatment group. The GWM damage index for each test group was calculated via the following formula:

$$
\begin{aligned}
\text{Damage index} (\%) = \Big[ &\sum (\text{number of damaged combs at each grade} \\
&\times \text{representative value of the specific grade})/ \\
&(\text{total number of combs from test colonies} \\
&\times \text{representative value of the highest damage grade}) \Big] \times 100.
\end{aligned}
$$

$$
\begin{aligned}
\text{Control efficacy} (\%) = \Big[ &(\text{damage index in control group/lure control group} \\
&- \text{damage index in lure} + \text{BiotGm group})/ \\
&\text{damage index in control group/lure control group} \Big] \times 100.
\end{aligned}
$$

### Statistical analysis, reproducibility, and visualization

Details of the statistical test, *P*-value indicating significance, and number of biological replicates for each experiment are shown in the figure legends. The arcsine square-root (sqrt) transformation, two-sided unpaired Student's *t*-test, two-sided Mann–Whitney *U*-test, two-sided chi-square test, Kruskal–Wallis test, and Kaplan–Meier survival analysis with the log-rank test were conducted using SAS 9.4 software (SAS Institute Inc, Cary, NC, USA). All experiments were performed with at least three or more biological replicates. Data were visualized using Origin 2021 software (Origin labs, USA) and ArcGIS Desktop 10.2 (ESRI, Redlands, CA, USA).

**Reporting summary**

Further information on research design is available in the Nature Portfolio Reporting Summary linked to this article.

## Data availability

All data supporting the findings of this study are available within the manuscript and its Supplementary Information files. The genome of *Bt* BiotGm was sequenced and the data were deposited in NCBI at Bio-Project PRJNA998231 (raw sequence data under accession number SRX22206352, assembly sequences under accession numbers CP130743 to CP130747). The web-based tool·the *Bacillus* Typing Bioinformatics Database[67] used in the CVTree phylogenetic analysis can be obtained from https://btbidb.com. US Environmental Protection Agency's BeeREX model data used in this study could be downloaded from https://www.epa.gov/sites/production/files/2015-11/beerexv1.0.xlsx. Exact *P*-values are included within the Source Data file as well. Source data are provided with this paper.

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

## Acknowledgements

We thank Pinhong Wang, Guirong Li, and Yujuan Qiu (Institute of Apicultural Research, China Academy of Agricultural Sciences) for beekeeping. We thank students Huali Li for participating in heat stability test, Tianxin Tang for participating in the safety test of BiotGm on honey bees, Sa Yang for participating in the histopathology test, Yuxin Kang and Hongyang Zhao for data collection of the field test. We thank Qihua Luo from Miyun Bureau of Landscape and Forest, Yonggui Zhang, Chongbo Liang and Xihong Fang from Beijing Landscaping Industry Promotion Center, Xing Wang from Beijing Apicultural Station, Wuguang Ye, Weiliang Zhou, Chuanlian Zhang, Zhuolin Xiong and Zheyun Zhao from Jiangxi Institute of Apicultural Research, Xingjiang He from Modern Agricultural Development Institute of Guizhou Academy of Agricultural Sciences, Shuang Yang from Sericultural and Apicultural Research Institute of Yunnan Academy of Agricultural Sciences, Yong Li from Guyuan Apicultural Station, Mao Feng from Institute of Apicultural Research, China Academy of Agricultural Sciences, and beekeepers (Li Chen, Xiaoli Guo, Caitai Duan, Fenhua Leng, Runsheng Yuan, Jinliang Liu, Shihong Zhou, Yuanhui Zhang and Jingyun Li) for testing the field biocontrol efficacy of entrapment against greater wax moth in Chinese

honey bee colonies. This work was supported by the National Key Research and Development Program (2022YFD1600202, 2021YFD1100309, P.D.), Agricultural Science and Technology Innovation Program (CAAS-ASTIP–2021-IAR, P.D.), and Central Public-interest Scientific Institution Basal Research Fund (mfsywf2020-01, P.D).

## Author contributions

Conceptualization: B.H., L.G., J.Z., P.D.; Formal analysis: B.H., L.Z., J.G., P.D.; Funding acquisition: P.D.; Investigation: L.G., J.W., J.G., T.W., Y.L., L.K., F.L., H.S., X.W., S.M., H.Z., Y.W., Y.L., Q.W., Q.D., P.D.; Methodology: B.H., L.Z., L.G., J.W., A.L., T.W., Y.L., L.K., F.L., H.S., S.M., P.D.; Data collection: Y.W., Y.L., F.L., H.S., H.Z., S.M., P.D.; Validation: B.H., L.Z., L.G., H.J., J.Z., P.D.; Visualization: B.H., L.Z., H.J., J.G., P.D.; Writing—original draft: B.H., L.Z., L.G., H.J., P.D.; Writing—review and editing: B.H., L.Z., L.G., H.J., J.Z., P.D.

## Competing interests

The authors declare no competing interests.
