## [Peer Review File · Nature Communications]

Reviewers' Comments:

Reviewer #1:

Remarks to the Author:

The basic goal of this study is to develop novel formulations and application methods for the bioinsecticide *Bacillus thuringiensis*. This is important from an applied, problem-solving perspective. I did not, however, see in the introduction a broader, more conceptual framework for the study. Does this work address any more fundamental questions relevant to bioinsecticides that can find more general application?

The authors report a lot of work here – this is a comprehensive and impressive research effort. However, I do have some concerns. The most important of these is that, contrary to the text of the paper, there is no field test of the efficacy of the control device presented here. I also thought this paper would describe the development of the novel Bt strain (BiotGm), but that work is not reported here.

Some suggestions for the authors to consider:

Line 77. BiotGm has not yet been introduced in the main text; please explain to the reader here what it is.

Lines 401-2. It's fine to refer the reader to another paper for the methodological details of the bioassays for honeybees; but here I would recommend at least a brief summary of the methods.

Line 419. Is the "flour" noted here wheat flour?

Line 458. Please explain what the acronym TUNEL stands for when it is first introduced.

Figure 1: This information is probably not necessary here. I think it would suffice to say that this pest is distributed worldwide.

Figure 4d. The authors write that BiotGm has no significant 'overall' effect on daily syrup and pollen consumption by adult bees (*A. mellifera* and *A. cerana*), but the data suggest a different story. First, I would recommend that Figure 4d (just this panel) be re-drawn so that time is on the x-axis and food consumption on the y-axis. I would also recommend that all four sub-panels be shown with axes drawn in, so that the reader can see the magnitude of the effect sizes. For both bee species and both foods, there is a clear trend for reduced food consumption in the presence of BiotGm. On some days the difference is statistically significant, on other days not, but the overall trend seems quite clear. In some cases, the effect size appears to be substantial (up to a doubling of the amount of food consumed). This is the only hint of negative effects of the biopesticide on the bees; I think it needs to be acknowledged.

(At some point in the manuscript, it would be helpful to define "dark comb".

Although the authors state that their entrapment device + lure + BiotGm was tested in the field, as far as I can see from the main text and supplementary text, figures, and tables, in fact there was no real test, because there appears to have been no control group (there was only the BiotGm treatment). Without a control, we simply have no way to know if the control device was effective, or if the pest simply was not present at these locations at this time. No data are presented in this paper to demonstrate the background level of infestation of commercial hives by *Galleria* moths in the region where this research was conducted. In my part of the world, *Galleria* can be a pest, but it is very rare; I have no idea how common it is in China.

The manuscript needs a final light editing to correct numerous errors with the English grammar.

Reviewer #2:

Remarks to the Author:

This paper describes a novel methodology for the control of the wax moth, *Galleria mellonella*, an important pest of honeybee colonies worldwide, which combines the use of a strain of *Bacillus thuringiensis* with high levels of selectivity against wax moth larvae with a novel lure-based entrapment device. The manuscript needs some revision to be considered for publication.

General comment: the authors need to improve the consistency of English in the manuscript (e.g. see comments on Abstract).

Title: the title does not adequately describe the control method developed and should be revised.

Abstract: some improvements in the English are required, e.g. line 21 - change 'their' to 'its'; line 25 - change 'generated' to 'developed', include 'method' after 'entrapment', and change the next part of the sentence to 'consisting of a highly effective lure for GWM, BiotGM, and a trapping device for GWM that prevents bees contacting the lure, which was found.....'; line 28 -0 suggest delete 'compelling'.

Introduction: lines 40-41: suggest ending sentence with '...food production'[and delete rest of sentence]; lines 42-43 - this sentence needs revising since it states that disease is the most significant contributor to the decline in bee colonies and does not mention pests, one of which, GWM, is the subject of the paper; lines 56-63 - this section (on Bt) needs rewriting, e.g. the sentence starting 'This bacterium..' is unnecessary and could be deleted; lines 83-85 - the final sentence could be deleted.

Results: delete lines 87-103 including Fig. 1 -'The damage status of the GWM' is unnecessary in the results section of this paper, the importance of GWM has already been mentioned in the introduction; lines 105-106 - delete sentence since this is part of the Methods; lines 106-108 - rewrite this sentence, removing the LC50 values since they are given in Table 1 and combine with the next sentence; lines 109-110 - change to 'was no significant difference ($P>0.01$)...' [Methods section should note and reference the significance level when comparing 95% CIs]; line 115 - change to 'Fig. 1'; lines 126-127 - change to Fig. 1; delete part a of Figure (now Fig. 1), since this is a method; lines 130-135 - change Fig. legend, deleting first part on bioassay; lines 140, 147, 154, 159, 161 - change to Fig. 2.; lines 187, 193, 196, 198 - change to Fig. 3.; lines 216-218 - suggest combine first two sentences deleting the Fig. reference for the olfactometer as this is method, e.g. 'Our olfactometer studies revealed that... (Fig. 4); Line 227 - change to Fig. 4 and delete 'part a' of the Figure and reference to it in lines 227-228; lines 238, 241, 243, 245, 247, 253 - change to 'Fig. 5..'.

Discussion: the discussion should be edited to remove undue repetition of the Introduction and Results text; lines 279, 284 - delete references to 'Fig. 1..'; line 298 - no need to include the LC50 value; line 302 - change to 'Fig. 2'; line 332 - change to 'Fig. 3'; line 349 - change to 'Fig. 2'; line 365 - change 'Fig. 4'; line 370 - change to 'Fig. 5'.

Methods: lines 403-415 - delete this section; line 434 - delete 'Fig. 2a'; line 555 - delete 'Fig. 5a'; lines 583, 588 change to 'Fig. 5a'.

Dear Editor,

We would like to express our deepest gratitude to both reviewers for their thorough and insightful comments and suggestions on our manuscript. The constructive feedback has significantly enhanced the quality and clarity of our work. Below, we have addressed the reviewer's concerns point by point:

Reviewer #1 (Remarks to the Author):

*1. The basic goal of this study is to develop novel formulations and application methods for the bioinsecticide *Bacillus thuringiensis*. This is important from an applied, problem-solving perspective. I did not, however, see in the introduction a broader, more conceptual framework for the study. Does this work address any more fundamental questions relevant to bioinsecticides that can find more general application?*

Response: Thank you for your constructive comments and feedback, which have greatly helped in refining our manuscript. We are grateful for pointing out the omission in providing a broader, conceptual framework. In the revised manuscript, we have enhanced the introduction to encapsulate the core challenges in the bioinsecticide domain (Refer to **Lines 57-67**). We aim to underline the relevance and potential of our research in offering solutions to these challenges.

2. The authors report a lot of work here – this is a comprehensive and impressive research effort. However, I do have some concerns. The most important of these is that, contrary to the text of the paper, there is no field test of the efficacy of the control device presented here.

Response: We regret any misunderstanding from our original manuscript regarding the field test. Indeed, field tests were conducted in chosen commercial breeding grounds for *Apis cerana cerana* with a documented history of severe wax moth invasion. Our objective was to assess the prospective field effectiveness of our novel biocontrol entrapment. We have elucidated the comparison to historical data and initial determination of efficacy (Refer to **Supplementary Fig. 6, 7** and **Supplementary Table 7**).

Upon receiving and considering the expert's feedback during the manuscript revision process, we took immediate action. To further validate our findings and address the reviewer's concerns, we initiated systematic field trials at *Apis cerana cerana* apiaries in July of this year, and these trials have been ongoing for over two months. We sincerely value the reviewer's guidance which prompted us to conduct these additional, rigorous tests. These trials are systematic, involving control groups, and offer robust evidence supporting the efficacy of our biocontrol entrapment against the greater wax moth. We've updated the manuscript with comprehensive details on these trials and their conclusive results, ensuring clearer understanding (Refer to **Lines 253-309**, and the accompanying **Fig. 7** and **Supplementary Fig. 20**).

Fig. 7 Field biocontrol efficacy of entrapment against GWMs.

Supplementary Figure 20. Photographs of three *A. cerana cerana* apiary field sites.

3. I also thought this paper would describe the development of the novel *Bt* strain (*BiotGm*), but that work is not reported here.

Response: We apologize for the oversight in omitting a thorough introduction to the *BiotGm* strain. To rectify this, we have incorporated comprehensive details about *BiotGm* in the Supplementary Results 1 section. *BiotGm*, isolated by our research team in China, possesses substantial potential for pest management. Its genome has been sequenced and deposited in GenBank. For a deeper insight, we provided a breakdown of its genetic components and its phylogenetic classification, and

have added visual evidence through images of BiotGm's crystals (accompanying **Supplementary Fig. 13** and **Supplementary Tables 8, 9**).

Supplementary Figure 13. Images of crystals and CVTree phylogenetic analysis for BiotGm.

Supplementary Table 8. The sequence features of *B. thuringiensis* strain BiotGm genome.

Name	GenBank accession number	Bases	GC (%)	CDS	Gene average length (bp)
Chr1	CP130743	5,718,068	35.28	6017	768
Plasmid1	CP130744	410,916	32.71	370	790
Plasmid2	CP130745	295,914	33.05	283	738
Plasmid3	CP130746	85,817	30.80	111	654
Plasmid4	CP130747	72,113	32.35	88	639

Supplementary Table 9. Annotation results of insecticidal protein of *B. thuringiensis* strain

BiotGm.

Protein Name	Length (bp)	Location	Annotation (GenBank No)	Amino acid similarity (%)
Cry1Aa1	3393	Plasmid2_00247	Cry1Aa1 (AAA22353.1)	99.99
Cry1Ca7	3570	Plasmid1_00270	Cry1Ca7 (AAG50438.1)	100.00
Cry1Da1	3498	Plasmid1_00266	Cry1Da1 (CAA38099.1)	100.00
Cry1Ia10	2160	Plasmid2_00248	Cry1Ia10 (AAP86782.1)	100.00
Cry2Ab1	1902	Plasmid2_00239	Cry2Ab1 (AAA22342.1)	100.00
Cry9Ea1	3453	Plasmid2_00254	Cry9Ea1 (BAA34908.1)	100.00
Vip3Aa11	2370	Plasmid2_00236	Vip3Aa11 (AAR36859.1)	100.00

We also appreciate the detailed suggestions provided in the subsequent section of the reviewer's comments. We have addressed each point individually to ensure clarity and have made respective amendments in the manuscript:

Some suggestions for the authors to consider:

1. Line 77. *BiotGm* has not yet been introduced in the main text; please explain to the reader here what it is.

Response: We appreciate the feedback and have added more details about the BiotGm strain and its significance in the revised manuscript (Refer to **Lines 83-85** and **Lines 432-435**). Moreover, we also included more information about the characterization of the BiotGm strain in the Supplementary Results section 1 (Refer to **Supplementary Fig. 13** and **Supplementary Tables 8, 9**).

2. Lines 401-2. It's fine to refer the reader to another paper for the methodological details of the bioassays for honeybees; but here I would recommend at least a brief summary of the methods.

Response: Thank you for highlighting this gap. We have now summarized the methods in the main

text for readers' convenience (Refer to **Lines 446-459, 563-579, 598-600**).

3. *Line 419. Is the “flour” noted here wheat flour?*

Response: We've clarified the reference to flour as wheat flour as recommended (Refer to **Line 463**).

4. *Line 458. Please explain what the acronym TUNEL stands for when it is first introduced.*

Response: In the revised version, we have provided the full definition of TUNEL when it is first introduced (Refer to **Lines 114-115**).

5. *Figure 1: This information is probably not necessary here. I think it would suffice to say that this pest is distributed worldwide.*

Response: We concur with the reviewer's suggestion and have moved the information to the Supplementary information section (Refer to **Supplementary Fig. 14**), emphasizing the importance of GWM as a global honeybee pest (Refer to **Lines 312-320**).

6. *Figure 4d. The authors write that BiotGm has no significant ‘overall’ effect on daily syrup and pollen consumption by adult bees (*A. mellifera* and *A. cerana*), but the data suggest a different story. First, I would recommend that Figure 4d (just this panel) be re-drawn so that time is on the x-axis and food consumption on the y-axis. I would also recommend that all four sub-panels be shown with axes drawn in, so that the reader can see the magnitude of the effect sizes. For both bee species and both foods, there is a clear trend for reduced food consumption in the presence of BiotGm. On some days the difference is statistically significant, on other days not, but the overall trend seems quite clear. In some cases, the effect size appears to be substantial (up to a doubling of the amount of food consumed). This is the only hint of negative effects of the biopesticide on the bees; I think it needs to be acknowledged.*

Response: We sincerely appreciate your in-depth review and agree with your observations. The figure has been revised as recommended (Refer to **Line 191, Fig. 4d**), and we've added acknowledgment of the potential negative effects in the text (Refer to **Lines 185-190**).

7. *At some point in the manuscript, it would be helpful to define “dark comb”.*

Response: In the revised manuscript, we have provided a definition for "dark comb" upon its first mention (Refer to **Line 211**).

8. Although the authors state that their entrapment device + lure + BiotGm was tested in the field, as far as I can see from the main text and supplementary text, figures, and tables, in fact there was no real test, because there appears to have been no control group (there was only the BiotGm treatment). Without a control, we simply have no way to know if the control device was effective, or if the pest simply was not present at these locations at this time.

Response: We recognize the importance of having a control group in our trials. We have addressed this issue and provided updated results in the revised manuscript (Refer to **Lines 253-309**, and the accompanying **Fig. 7** and **Supplementary Fig. 11**).

Supplementary Figure 11. Representative images of the attractive effect and control efficacy of entrapment in field conditions.

9. No data are presented in this paper to demonstrate the background level of infestation of commercial hives by *Galleria* moths in the region where this research was conducted. In my part of the world, *Galleria* moths can be a pest, but it is very rare; I have no idea how common it is in China.

Response: Thank you for this observation. We've now included detailed data from our monitoring efforts over the past years, as you've mentioned. It should help provide a clearer understanding of the pest's significance in the context of Chinese honey bee colonies (Refer to **Lines 314-316**, and the accompanying **Supplementary Table. 10**).

Supplementary Table 10. Monitoring of the occurrence of *Galleria mellonella* in the main breeding areas of *A. cerana cerana* in China in recent years

No.	Beekeepers	Address	Survey year	Occurrence of greater wax moth
1	Xiaoli Guo	Miyun, Beijing (40°23'N, 116°46'E)	2020, 2021	[x] Yes [ ] No
2	Pinghong Wang	Shijingshan, Beijing (39°58'N, 116°8'E)	2017-2021	[x] Yes [ ] No
3	Runsheng Yuan	Jinzhong, Shanxi province (37°25'N, 112°34'E)	2019-2021	[x] Yes [ ] No
4	Jingyun Li	Longnan, Gansu province (33°46'N, 106°4'E)	2019-2021	[x] Yes [ ] No
5	Yong Li	Guyuan, Ningxia province (36°0'N, 106°14'E)	2020, 2021	[x] Yes [ ] No
6	Shihong Zhou	Ningshan, Shaanxi province (33°23'N, 108°17'E)	2019-2021	[x] Yes [ ] No
7	Yuanhui Zhang	Puge, Sichuan province (27°22'N, 102°32'E)	2019-2021	[x] Yes [ ] No
8	Shilong Ma	Enshi, Hubei province (30°19'N, 109°28'E)	2019-2021	[x] Yes [ ] No
9	Fenghua Leng	Xiushui, Jiangxi province (29°3'N, 114°9'E)	2017-2021	[x] Yes [ ] No
10	Li Chen	Ziyun, Guizhou province (26°30'N, 106°39'E)	2019-2021	[x] Yes [ ] No
11	Caitai Duan	Chengbu, Hunan province (26°23'N, 110°19'E)	2019-2021	[x] Yes [ ] No

10. *The manuscript needs a final light editing to correct numerous errors with the English grammar.*

Response: We have enlisted the services of the professional editing agency *American Journal Experts* to refine our manuscript, ensuring its clarity and coherence. A proofreading certificate from the agency has been attached.

Reviewer #2 (Remarks to the Author):

*This paper describes a novel methodology for the control of the wax moth, *Galleria mellonella*, an important pest of honeybee colonies worldwide, which combines the use of a strain of *Bacillus thuringiensis* with high levels of selectivity against wax moth larvae with a novel lure-based entrapment device. The manuscript needs some revision to be considered for publication.*

General comment:

1. *the authors need to improve the consistency of English in the manuscript (e.g. see comments on Abstract).*

Response: We are grateful for your detailed feedback on the manuscript's English consistency. We've thoroughly reviewed the entire document and made the necessary corrections, especially in the abstract, to ensure clarity and coherence.

2. *Title: the title does not adequately describe the control method developed and should be revised.*

Response: Thank you for your recommendation. We've adjusted the title to better reflect the

methodology described in the manuscript. The revised title is: “**Green control of the greater wax moth by integrating *Bacillus thuringiensis* and entrapment for global apiculture**”.

Abstract:

3. some improvements in the English are required, e.g. line 21 - change 'their' to 'its'; line 25 - change 'generated' to 'developed', include 'method' after 'entrapment', and change the next part of the sentence to 'consisting of a highly effective lure for GWM, BiotGM, and a trapping device for GWM that prevents bees contacting the lure, which was found.....'; line 28 -0 suggest delete 'compelling'.

Response: We value your suggestions for improving the abstract. All recommended edits have been incorporated to enhance clarity and coherence.

Introduction:

4. lines 40-41: suggest ending sentence with '...food production'[and delete rest of sentence]; lines 42-43 - this sentence needs revising since it states that disease is the most significant contributor to the decline in bee colonies and does not mention pests, one of which, GWM, is the subject of the paper; lines 56-63 - this section (on Bt) needs rewriting, e.g. the sentence starting 'This bacterium..' is unnecessary and could be deleted; lines 83-85 - the final sentence could be deleted.

Response: We appreciate your insights regarding the Introduction section. Based on your guidance:

- (1) We've truncated the sentence on lines 40-41 for precision (Refer to **Lines 44-45**).
- (2) Revised lines 42-43 to encompass both disease and pests as contributors to the decline in bee colonies (Refer to **Lines 46-48**).
- (3) Condensed the section on Bt (lines 56-63) by removing redundant information (Refer to **Lines 57-67**).
- (4) The final sentence on lines 83-85 has been deleted for clarity.

Results:

5. delete lines 87-103 including Fig. 1 - 'The damage status of the GWM' is unnecessary in the results section of this paper, the importance of GWM has already been mentioned in the introduction; lines 105-106 - delete sentence since this is part of the Methods; lines 106-108 - rewrite this sentence, removing the LC50 values since they are given in Table 1 and combine with the next sentence; lines 109-110 - change to 'was no significant difference ($P > 0.01$)...' [Methods section should note and reference the significance level when comparing 95% CIs]; line 115 - change to 'Fig. 1'; lines 126-127 - change to Fig. 1; delete part a of Figure (now Fig. 1), since this is a method; lines 130-135 - change Fig. legend, deleting first part on bioassay; lines 140, 147, 154, 159, 161 - change to Fig. 2.; lines 187, 193, 196, 198 - change to Fig. 3.; lines 216-218 - suggest combine first two sentences deleting the Fig. reference for the olfactometer as this is method, e.g. 'Our olfactometer studies revealed that... (Fig. 4); Line 227 - change to Fig. 4 and delete 'part a' of the Figure and reference

to it in lines 227-228; lines 238, 241, 243, 245, 247, 253 - change to 'Fig. 5..!'

Response: Your detailed guidance on the Results section is appreciated. We've made the following changes in line with your recommendations:

- (1) Removed the original Fig. 1 and specified lines for "the damage status of the GWM" in the results, discussion and methods sections.
- (2) The sentence on lines 105-106 has been deleted as it belongs in the Methods.
- (3) Lines 106-108 and 109-110 have been revised for clarity, with LC₅₀ values are removed, and the significance level was referenced when comparing 95% CIs (Refer to **Lines 94-96**).
- (4) Regarding your suggestion on Figure 2a: We acknowledge your point about the methodological nature of this flowchart and its initial pairing with Figure 2b. The flowchart, however, encapsulates our innovative bioassay technique, which is directly linked to the results displayed in original Table 1. This technique, elaborated on **lines 333-349**, fills an existing knowledge gap and offers pivotal insights into the nutritional dynamics of GWM larvae.

To more clearly depict the integral relationship between the methodology and the findings, we've decided to merge Figure 2a with Table 1. This consolidated representation will be presented as Figure 1 in our revised manuscript. We believe that this new arrangement not only provides a cohesive understanding of our approach but also accentuates the importance of our method in the context of the study's results. We hope this adaptation addresses your concerns while emphasizing the core essence of our research.

- (5) The sentence on lines 216-218 have been combined and revised for clarity (Refer to **Lines 210-212**), and have been deleted the Fig. reference (now **Fig. 5a**) and the specified lines 227-228 for the olfactometer as it belongs in the Methods.
- (6) The order and referencing of figures have been adjusted throughout this section in response to the removal of some figures. All textual descriptions linked to these figures have also been updated.

Discussion:

6. the discussion should be edited to remove undue repetition of the Introduction and Results text;

Response: We concur with the reviewer's input and have edited the beginning of the discussion to eliminate redundancy. The revised discussion now delves directly into the broader implications and context of our findings, providing a more comprehensive understanding of the study's importance.

7. lines 279, 284 - delete references to 'Fig. 1..!'; line 298 - no need to include the LC50 value; line 302 - change to 'Fig. 2!'; line 332 - change to 'Fig. 3!'; line 349 - change to 'Fig. 2!'; line 365 - change 'Fig. 4!'; line 370 - change to 'Fig. 5!'

Response: Your detailed guidance on the Discussion section is appreciated. We've made the following changes in line with your recommendations:

- (1) References to the respective figures have been updated as advised.
- (2) The LC₅₀ value (line 298) have been deleted as it is *no need* in the Discussion.

Methods:

8. lines 403-415 - delete this section; line 434 - delete 'Fig. 2a'; line 555 - delete 'Fig. 5a'; lines 583, 588 change to 'Fig. 5a'.

Response: We concur with the reviewer's input on the irrelevance of certain sections in the Methods. They have been removed or updated accordingly.

Thank you once again for your thoughtful insights, which have undeniably enriched our work. We believe that with these revisions, our manuscript now offers a more comprehensive and clearer understanding of our research.

Warm regards,

Pingli Dai

Reviewers' Comments:

Reviewer #1:

Remarks to the Author:

The authors have done an excellent job in responding to reviewer suggestions. In particular, the authors have conducted a full experimental test of the efficacy of the lure + Bt pathogen in the field that included control treatments. This field experiment provides strong evidence for the efficacy of the novel control system under field conditions.

Dear Editor,

We would like to express our deepest gratitude to both reviewers for their thorough and insightful comments and suggestions on our manuscript. The constructive feedback has significantly enhanced the quality and clarity of our work. Below, we have addressed the reviewer's concerns point by point:

Reviewer #1 (Remarks to the Author):

The authors have done an excellent job in responding to reviewer suggestions. In particular, the authors have conducted a full experimental test of the efficacy of the lure + Bt pathogen in the field that included control treatments. This field experiment provides strong evidence for the efficacy of the novel control system under field conditions.

Response: Thanks a lot for the largely positive comments made by you on our study.

Thank you once again for your thoughtful insights, which have undeniably enriched our work.

Warm regards,

Pingli Dai